# Strat-Reasoner: Reinforcing Strategic Reasoning of LLMs in Multi-Agent Games

Yidong He [* 1]  Yutao Lai [* 1]  Pengxu Yang [1]  Jiarui Gan [2]  Jiexin Wang [1]  Yi Cai [1]  Mengchen Zhao [1]

## Abstract

While Large Language Models (LLMs) excel in certain reasoning tasks, they struggle in multi-agent games where the final outcome depends on the joint strategies of all agents. In multi-agent games, the non-stationarity of other agents brings significant challenges on the evaluation of the reasoning process and the credit assignment over multiple reasoning steps. Existing single-agent reinforcement learning (RL) approaches and their multi-agent extensions fail to address these challenges as they do not incorporate other agents in the reasoning process. In this work, we propose Strat-Reasoner, a novel RL-based framework that improves LLMs' strategic reasoning ability in multi-agent games. We introduce a novel recursive reasoning paradigm where an agent's reasoning also integrates other agents' reasoning processes. To provide effective reward signals for the intermediate reasoning sequences, we employ a centralized Chain-of-Thought (CoT) comparison module to evaluate the reasoning quality. Finally, we compute an accurate hybrid advantage and develop a group-relative RL approach to optimize the LLM policy. Experimental results show that Strat-Reasoner substantially improves strategic abilities of underlying LLMs, achieving 22.1% average performance improvements across various multi-agent games. Code is publicly available at https://github.com/ydhe1012/Strat-Reasoner.

## 1. Introduction

Large Language Models (LLMs) have demonstrated remarkable proficiency in tasks requiring logical reasoning such as code generation (Achiam et al., 2023; Chen et al.) and mathematical reasoning (Guo et al., 2025). However, real-world scenarios such as Poker games and diplomatic negotiation, often involve strategic reasoning over other intelligent agents (Zhang et al., 2024). Current LLMs, even Large Reasoning Models (LRMs), perform poorly in multi-agent environments because single-agent reasoning paradigm is non-strategic. Therefore, it is urgent to improve LLMs' strategic reasoning ability from a multi-agent perspective.

Reinforcement Learning (RL) has achieved notable success in enhancing LLM reasoning (Ouyang et al., 2022; Rafailov et al., 2023). However, in the context of LLM agents, extending single-agent RL to the multi-agent setting faces three critical challenges. First, it is hard to address the uncertainty of other agents. In conventional multi-agent RL (MARL) works, the uncertainty of other agents is usually reduced by joint policy optimization and opponent modeling (Busoniu et al., 2008). However, LLM agents are more unpredictable as their decision making usually involves a complex reasoning process (Zhao et al., 2024). Second, there is no effective signal to guide the LLM agent's reasoning process. While conventional MARL addresses the credit assignment problem between agents, the decomposed rewards are far from enough to guide the reasoning of LLM agents (Liao et al., 2025). Third, in the multi-turn setting, each agent accumulates multiple turns of related reasoning sequences, making it hard to evaluate the contribution of each reasoning process (Nguyen et al., 2018).

Recent works have attempted to extend Group Relative Policy Optimization (GRPO) (Guo et al., 2025), the predominant single-agent RL method, to multi-agent settings. Borrowing ideas from game theory, some approaches employ a self-play framework where agents share the same parameters (Liu et al., 2025; Yuan et al., 2025). However, lacking explicit modeling of opponents' reasoning, this self-play paradigm fails to intrinsically capture the non-stationarity introduced by unknown opponents. Other works integrate GRPO into cooperative MARL frameworks, relying on timestep-aligned average returns as baselines (Liu

*Equal contribution [1]School of Software Engineering, South China University of Technology [2]Department of Computer Science, University of Oxford. Correspondence to: Mengchen Zhao <zzmc@scut.edu.cn>.

*Proceedings of the 43rd International Conference on Machine Learning*, Seoul, South Korea. PMLR 306, 2026. Copyright 2026 by the author(s).

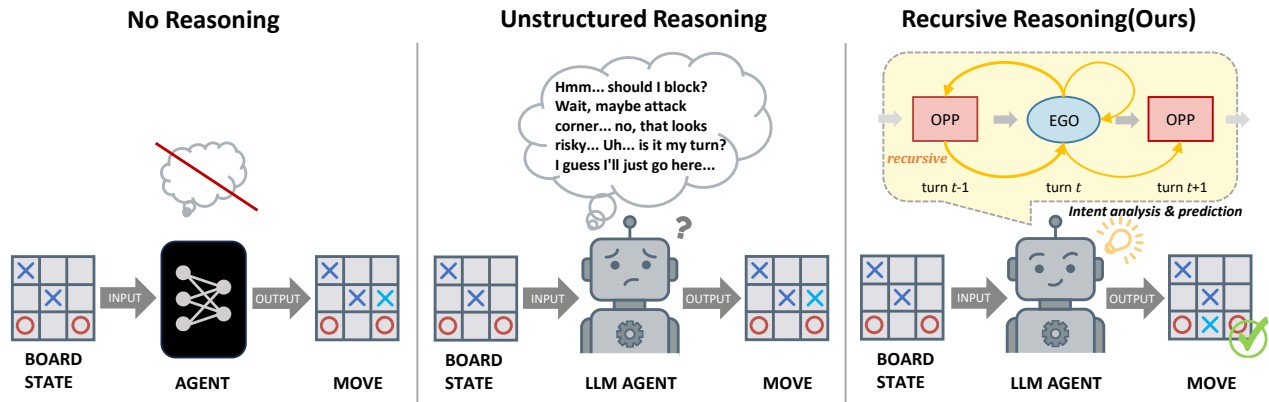

*Figure 1.* Comparison of reasoning paradigms in strategic decision-making. Unlike **No Reasoning** (Left) and **Unstructured Reasoning** (Middle) which fail to handle complex strategic traps, our **Recursive Reasoning** paradigm (Right) employs a structured, multi-step reasoning process. By explicitly reasoning about the opponent's intent and predictions in a recursive way, our method achieves superior strategic performance, as demonstrated by the successful move, and interpretability, which enables intermediate training signals.

et al., 2026). However, the divergence of long-horizon trajectory renders late-game states incomparable, causing severe baseline variance and unreliable advantage estimation. Furthermore, these existing works inherit GRPO's reliance on response-level rewards, consequently lacking the fine-grained feedback necessary to guide reasoning steps.

To address these limitations, we propose **Strat-Reasoner**, an RL-based framework designed to enhance strategic reasoning for LLMs in multi-agent games. Strat-Reasoner utilizes opponent modeling in the context of LLMs by explicitly integrating the opponents' reasoning process into the ego agent's decision-making. Intuitively, this mimics the human cognitive capability of "thinking about what others think." Strat-Reasoner consists of three key modules. **First**, drawing ideas from the hierarchical cognitive framework (Camerer et al., 2004) in game theory, the Recursive Reasoning module structures each agent's reasoning as a recursive process that explicitly considers other agents' beliefs. Figure 1 illustrates the differences between the recursive reasoning and conventional reasoning paradigms. **Second**, the Centralized CoT Comparison module computes reliable CoT scores for each agent by measuring the alignment between the ego agent's prediction and the opponent's actual reasoning process. **Third**, to mitigate long-horizon stochasticity, the Hybrid Advantage Estimation module integrates intermediate CoT scores with conventional return-based advantages, thereby achieving reliable advantage estimation.

Our contributions can be summarized as follows:

- Inspired by game theory, we propose a novel recursive reasoning paradigm that incentivizes strategic behaviors of LLMs in multi-agent games.

- We develop Strat-Reasoner, a novel RL framework that integrates centralized CoT comparison and hybrid

advantage estimation to enhance the strategic reasoning capability of LLMs. These designs ensure fine-grained credit assignment over multiple reasoning sequences.

- We conduct extensive experiments across both competitive and cooperative games, validating that Strat-Reasoner incentivizes superior strategic reasoning capabilities with robust generalization to unseen games.

## 2. Related Works

**Chain-of-Thought incentivized reasoning.** Chain-of-Thought (CoT) enables Large Language Models (LLMs) to generate intermediate rationales, significantly improving their reasoning capabilities (Wei et al., 2022). While zero-shot approaches (Kojima et al., 2022) and self-consistency strategies (Wang et al.) have established the feasibility and robustness of this technique, subsequent works have extended the paradigm into structured tree reasoning (Yao et al., 2023). The synergy between reasoning and external tools, including action execution and self-correction (Yao et al., 2022; Chen et al.; Shinn et al., 2023), has facilitated the transition of LLMs toward autonomous agents. However, current efforts are largely confined to designing external workflows for the implicit utilization of CoT. There is still a lack of methods that fundamentally enhance an agent's intrinsic cognition and reasoning depth.

**Reinforcement learning for LLM reasoning.** Reinforcement learning is essential for enhancing the reasoning and instruction-following abilities of LLMs. While RLHF (Ouyang et al., 2022) and RLAIF (Bai et al., 2022) align models with human values via preference modeling, methods such as DPO (Rafailov et al., 2023) and GRPO (Shao et al., 2024) streamline the training process by optimizing reward structures. To improve reward objectivity, RLVR (Guo

et al., 2025; Wen et al., 2025) introduces verifiable external feedback. However, these existing approaches focus largely on single-turn, single-agent scenarios. They do not yet resolve the inherent sparse reward issues found in multi-turn, multi-agent environments, which are more representative of real-world applications.

**LLM-based multi-agent systems (LaMAS).** LaMAS (Guo et al., 2024) differs from the traditional MARL setting in the sense that LLM agents often move asynchronously and reason about each other. Current LaMAS-based studies predominantly focus on training-free approaches through workflow design (Du et al.; Wu et al., 2024). Although recent works like Multiagent Finetuning (Subramaniam et al., 2025) and MAPoRL (Park et al., 2025) have explored finetuning, they typically assume agent homogeneity—a simplification that limits their performance in diverse scenarios. Our approach enables agent learning in diverse games with heterogenous opponents, thereby incentivizing a more general strategic decision making capability.

# 3. Preliminaries

## 3.1. Problem Formulation

We focus on Two-player Alternating Markov Games (Littman, 1994), which covers a variety of applications such as board games, resource competition and cyber security. A Two-player Alternating Markov Game is formally defined by the tuple $\mathcal{G} = \langle \mathcal{S}, \mathcal{A}, P, r, \gamma \rangle$. To align with the generative nature of LLMs, the state space $\mathcal{S}$, representing the sequence of dialogue history, is partitioned into two disjoint sets $\mathcal{S}_1$ and $\mathcal{S}_2$, where Agent $i$ selects actions $a \in \mathcal{A}_i$ only when the current state $s \in \mathcal{S}_i$. We define the agent's generative output as a tuple $o = (z, a)$, where $z$ denotes the internal reasoning trace (e.g., chain-of-thought) and $a \in \mathcal{A}$ represents the observable action sent to the environment. The transition function $P : \mathcal{S} \times \mathcal{A} \to \mathcal{S}$ is typically deterministic in this context, such that the subsequent state is updated by appending the observable response, $s_{t+1} = [s_t; a_t]$. The reward function set $r = \{r_1, r_2\}$ specifies the immediate feedback $r_i : \mathcal{S} \times \mathcal{A} \to \mathbb{R}$ for each agent. To enforce a strictly alternating turn structure, the transition must satisfy the constraint that for any $s \in \mathcal{S}_i$ and $a \in \mathcal{A}_i$, $P(s, a, s') > 0$ implies $s' \in \mathcal{S}_j$ for $j \neq i$. Each agent $i \in \{1, 2\}$ aims to optimize its policy $\pi_i(o|s)$ to maximize the total return $R_i = \sum_{t=0}^{\infty} \gamma^t r_i(s_t, a_t)$.

While classical solvers excel in finite settings (v. Neumann, 1928; Watkins & Dayan, 1992), they struggle to solve the game $\mathcal{G}$ when the agents are LLMs. First, the space of generative outputs $o$ in LLM-based games which is rooted in natural language is effectively infinite, rendering traditional search or tabular methods intractable. Second, the LLM-specific dual structure $o = (z, a)$ introduces a unique credit

assignment challenge: it is difficult to discern whether a low reward $r$ stems from flawed internal reasoning $z$ or an ineffective external action $a$. Furthermore, the stochastic nature of LLMs implies that the mapping from reasoning $z$ to action $a$ is not deterministic, adding a layer of uncertainty that complicates policy optimization. This internal complexity makes solving $\mathcal{G}$ with LLMs far more challenging than traditional games with simple, atomic actions.

## 3.2. Group-Relative Policy Optimization

Group-Relative Policy Optimization (GRPO) is an efficient RL algorithm. To eliminate the need for a separate critic network (Schulman et al., 2017), GRPO simultaneously generate multiple responses $\{o^i\}_{i=1}^G$ and their corresponding rewards $\mathbf{r} = \{r^i\}_{i=1}^G$ for each question to calculate the group-relative advantage, which yields:

$$
\mathcal{J}(\theta) = \mathbb{E}_{q \sim P(Q), \{o^i\}_{i=1}^G \sim \pi_{\theta_{old}}}
$$
$$
\left[ \frac{1}{G} \sum_{i=1}^G \frac{1}{|o^i|} \sum_{t=1}^{|o^i|} \mathcal{J}_{surr}(\pi_\theta; \pi_{\theta_{old}}, A_t^i, \varepsilon) \right] \quad (1)
$$

$$
\mathcal{J}_{surr} = \min \left[ r_t^i(\theta) A_t^i, \text{clip}(r_t^i(\theta), 1 - \varepsilon, 1 + \varepsilon) A_t^i \right] \quad (2)
$$

where $r_t^i(\theta) = \frac{\pi_\theta(o_t^i | q, o_{<t}^i)}{\pi_{\theta_{old}}(o_t^i | q, o_{<t}^i)}$, $A_t^i = \frac{r^i - \text{mean}(\mathbf{r})}{\text{std}(\mathbf{r})}$.

For multi-turns scenario, we consider all $G$ trajectories as a group, i.e. $\{(s_k^i, a_k^i)_{k=1}^{K^i}\}_{i=1}^G$, $\mathbf{r} = \{R^i\}_{i=1}^G$ represents the terminal reward. To accommodate multi-turn interactions, the GRPO objective is extended by incorporating a summation across all turns:

$$
\mathcal{J}(\theta) = \mathbb{E}_{s_k^i \sim P(S), o_{k,t}^i \sim \pi_{\theta_{old}}(O | s_k^i, o_{k,<t}^i)}
$$
$$
\frac{1}{G} \sum_{i=1}^G \frac{1}{K^i} \sum_{k=1}^{K^i} \frac{1}{|o_k^i|} \sum_{t=1}^{|o_k^i|} \mathcal{J}_{\text{surr}}(\pi_\theta; \pi_{\theta_{\text{old}}}, A_{k,t}^i, \varepsilon) \quad (3)
$$

where $A_{k,t}^i = \frac{r^i - \text{mean}(\mathbf{r})}{\text{std}(\mathbf{r})}$.

# 4. Methods

**Motivation.** Strategic reasoning is essential for LLM agents to function effectively in multi-agent environments (Zhang et al., 2024). However, existing works predominantly optimize reasoning indirectly through final game outcomes (Yuan et al., 2025). In fact, in strategic multi-agent games, the reasoning process is the actual driver of complex behaviors and is highly sensitive to state variations. Moreover, the opponent's thought process contains rich strategic information that outcome-based signals fail to capture (Lightman et al., 2023). Based on the above insights, we believe that explicitly leveraging the opponent's reasoning process would significantly benefit agents in mastering strategic reasoning.

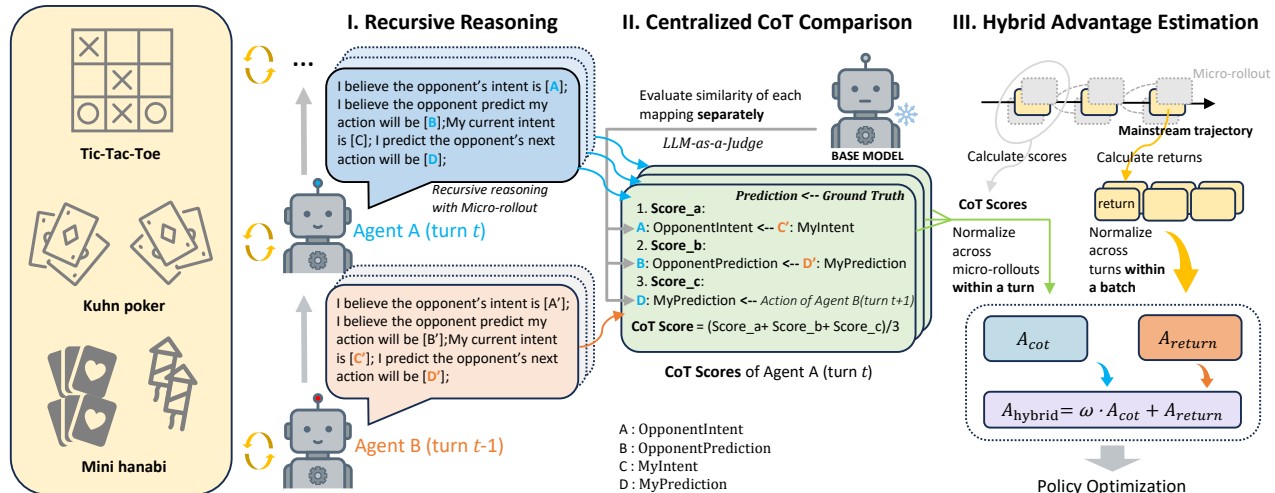

*Figure 2.* The overview of Strat-Reasoner framework. The diagram illustrates the policy optimization process for Agent A at turn $t$. Crucially, all of Agent A's micro-rollouts (blue bubble) are compared against Agent B's reasoning and actions in the mainstream trajectory (solid red bubble), rather than against Agent B's parallel micro-rollouts (dashed grey bubbles).

**Overview.** As illustrated in Figure 2, Strat-Reasoner operates in three synergistic phases: (I) Recursive Reasoning, where agents explicitly model the opponent's beliefs and intentions recursively. (II) Centralized CoT Comparison, which derives turn-level CoT scores by comparing the ego agent's predictions with the opponent's actual thoughts and actions. (III) Hybrid Advantage Estimation, which integrates the intermediate CoT scores with return-based advantages to facilitate stable and accurate policy optimization.

### 4.1. Recursive Reasoning in Two-player Alternating Markov Game

Strategic decision-making naturally demands the ability to model and predict the cognitive states of others. In classical game theory, this is formally characterized by Cognitive Hierarchy (CH) Theory (Camerer et al., 2004). This theory posits that intelligent agents do not act in isolation but operate at varying levels of strategic depth, which is typically embodied in practice through *recursive reasoning* (Gmytrasiewicz & Doshi, 2005), formalized as *"I believe that you believe that I believe..."* (Wen et al., 2019).

While this theoretical paradigm is well-established, effectively grounding it within LLMs for Two-player Alternating Markov Games (TAMG) requires a specialized design. In the TAMG setting, interactions are strictly sequential: the agent observes the opponent's move from the previous turn $(t-1)$, makes a decision at the current turn $(t)$, and anticipates the opponent's response in the next turn $(t+1)$, constituting a continuous alternating decision cycle. Direct application of abstract game theory often fails to capture the intricate interplay between strategic belief depth and fine-grained temporal dependencies, leading to reasoning

that is either too generic or disconnected from the specific game dynamics.

To bridge the gap between abstract theory and practical execution in TAMG, we design a customized Recursive Reasoning Module. Unlike generic CoT, this module is specifically structured to mirror the alternating nature of the game. We instantiate the recursive paradigm into a "Past-Present-Future" cognitive loop: the ego agent must first decode the opponent's intent from the past turn, then infer the opponent's prediction of the ego agent's current move, then formulate its own strategy, and finally predict the opponent's next move. This tailored structure ensures that the LLM's recursive reasoning is not just a linguistic template, but a strategic mechanism tightly coupled with the Markovian dynamics of the game.

By explicitly modeling the opponent, this recursive process not only deepens the agent's strategic understanding but also naturally implies mapping relations between the reasoning of the ego agent and the opponent. To facilitate fine-grained analysis and comparison, we refine the recursive reasoning segment into a structured reasoning block comprising four key fields, as illustrated in Figure 3:

1. **OpponentIntent**: The ego agent's belief about the opponent's **intent** in the previous turn (turn $t$-1).
2. **OpponentPrediction**: The ego agent's second-order belief, estimating the opponent's prediction of the ego agent's current **action** (turn $t$).
3. **MyIntent**: The ego agent's current **intent** (turn $t$).
4. **MyPrediction**: The ego agent's prediction of the opponent's **action** in the next turn (turn $t$+1).

To mitigate the potential cognitive degradation caused by

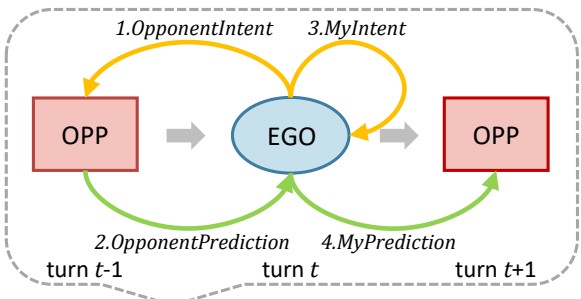

*Figure 3.* Illustration of the Recursive Reasoning structure. Yellow arrows represent intent-oriented reasoning (OpponentIntent, MyIntent), while green arrows denote action-oriented predictions (OpponentPrediction, MyPrediction).

rigid formatting and ordering constraints, we explicitly instruct the LLM to generate a free-form thinking process before synthesizing the structured reasoning block, which ensures the model can fully articulate its logic without being hindered by syntax requirements.

### 4.2. Centralized Comparison for CoT Score Computation

To address the challenge of sparse rewards, we argue that relying solely on formatting rewards is insufficient. While these signals ensure structural validity, they often lack correlation with the intrinsic game logic (Yuan et al., 2025). To provide meaningful strategic guidance, we draw upon the Centralized Training with Decentralized Execution (CTDE) paradigm (Lowe et al., 2017). Analogous to accessing invisible states in traditional MARL, we treat the CoT reasoning processes of agents as global information during centralized training. The core intuition is to evaluate the quality of the ego agent's reasoning by verifying whether its subjective beliefs align with the opponent's actual thoughts and actions.

Leveraging our structured reasoning block, we establish fine-grained mapping relations between the ego agent and the opponent. Specifically, for a reasoning process at turn $t$, we construct three key comparison pairs (as illustrated in Figure 2):

- **Past Alignment**: The ego agent's *OpponentIntent* (belief about the opponent's intent) is compared against the opponent's actual *MyIntent* at turn $t-1$.

- **Recursive Alignment**: The ego agent's *OpponentPrediction* (belief about the opponent's prediction of the ego agent's action) is compared against the opponent's actual *MyPrediction* at turn $t-1$.

- **Future Alignment**: The ego agent's *MyPrediction* (prediction of the opponent's next action) is compared against the opponent's actual action at turn $t+1$.

These three dimensions comprehensively evaluate the ego agent's strategic understanding of the past, recursive beliefs, and future outcomes.

To quantify this alignment, we employ an LLM-as-a-judge (Zheng et al., 2023) mechanism by directly leveraging the base model as the evaluator, thereby eliminating the need for external models or additional deployment overhead. By explicitly disentangling the reasoning process into independent semantic units, the evaluation of complex strategic reasoning is decomposed into literal comparisons between specific fields. This allows the judge to verify semantic similarity based solely on game rules without processing lengthy game contexts. Formally, the score for each component $k \in \{a, b, c\}$ at turn $t$ is defined as:

$$S_{k,t} = \text{Similarity}(\text{Prediction}_{k,t}, \text{GroundTruth}_{k,t}) \in [0, 1] \tag{4}$$

where $\text{GroundTruth}_{k,t}$ denotes the corresponding text retrieved from the opponent's reasoning in adjacent turns. The final turn-level **CoT Score** is the average of these components:

$$S_{\text{cot},t} = \frac{1}{3} \sum_{k \in \{a,b,c\}} S_{k,t} \tag{5}$$

It is important to note that $S_{cot,t}$ serves as an immediate evaluation signal rather than a cumulative reward. The quality of future reasoning does not necessarily reflect the merit of the current turn's action and reasoning; thus, crediting this score cumulatively would introduce bias. Instead, this score acts as a temporary signal to construct a variance-reduced advantage term, which will be elaborated in the next subsection.

### 4.3. Hybrid Advantage Estimation with Micro-rollout

While return-based optimization is standard in RL, relying solely on cumulative returns for long-horizon strategic games is problematic due to the high variance introduced by environmental stochasticity and the sparsity of intrinsic rewards. To address this, we propose a Hybrid Advantage mechanism that integrates the low-variance, immediate CoT signals with the unbiased, long-term returns.

**Micro-rollout for CoT Advantage.** To obtain a stable baseline for the immediate reasoning quality, we introduce the **Micro-rollout** technique. Specifically, at each decision turn $t$, the policy $\pi_\theta$ generates $M$ independent reasoning paths and actions in parallel:

$$\{(z_t^{(i)}, a_t^{(i)})\}_{i=1}^{M} \sim \pi_\theta(z_t, a_t|s_t) \tag{6}$$

Note that these micro-rollouts are strictly limited to the current time step for calculating CoT scores and do not proceed to future game branches (except for the primary rollout used

to continue the game interaction to generate mainstream trajectories). This design effectively decouples the dense reasoning evaluation from the expensive environment interaction, keeping the computational overhead minimal.

For engineering optimization, in practice, we avoid branching during the regular rollout phase. Instead, we perform an additional "resampling" phase in the pipeline after the rollout phase, where we collect the states as prompts for each turn and send them to the inference engine in batches, to generate $M$ micro-rollouts in parallel. This design enables Micro-rollouts of different turns to be generated concurrently, which significantly reduces the time overhead.

Based on $M$ generated responses, we calculate the turn-level **CoT Advantage** using GRPO normalization:

$$A_{\text{cot},t}^{(i)} = \frac{S_{\text{cot},t}^{(i)} - \text{mean}(\{S_{\text{cot},t}\})}{\text{std}(\{S_{\text{cot},t}\})} \quad (7)$$

where $S_{\text{cot},t}^{(i)}$ is the CoT score of the $i$-th micro-rollout at turn $t$ and $\{S_{\text{cot},t}\}$ denotes the set of CoT scores from all micro-rollouts at turn $t$.

The CoT Advantage serves as a variance-reduced proxy, providing dense feedback on the strategic reasoning quality independent of future stochasticity.

**Return-based Advantage.** To ensure the policy ultimately optimizes the game's intrinsic objective, we retain the standard return-based advantage. Following the design in MARSHAL (Yuan et al., 2025), for each turn, we perform a Monte Carlo summation of future rewards to obtain $R_t$.

To construct a stable baseline, we flatten the trajectories into individual turns and calculate the average return across the entire training batch (computed specifically for each agent role). The return-based advantage is then obtained by centering the cumulative return:

$$A_{\text{return},t} = R_t - \text{mean}(\{R_t\}_{\text{batch}}) \quad (8)$$

where $\{R_t\}_{\text{batch}}$ denotes the set of cumulative future returns from all samples in the current batch.

**Hybrid Optimization Objective.** The final advantage is a weighted combination of the two components, balancing the bias-variance trade-off:

$$A_{\text{hybrid},t} = A_{\text{return},t} + \omega \cdot A_{\text{cot},t} \quad (9)$$

where $\omega$ controls the weight of the reasoning signal.

Therefore, the optimization objective of Strat-Reasoner for each agent is:

$$\mathcal{J}(\theta) = \mathbb{E}_{s_k^i \sim P(S), o_{k,t}^{p,i} \sim \pi_{\theta_{\text{old}}}(O|s_k^i, o_{k,<t}^i)}$$

$$\frac{1}{G} \sum_{i=1}^{G} \frac{1}{K^i} \sum_{k=1}^{K^i} \frac{1}{|o_k^i|} \sum_{t=1}^{|o_k^i|} \mathcal{J}_{\text{surr}}(\pi_\theta; \pi_{\theta_{\text{old}}}, A_{\text{hybrid}}, \varepsilon) \quad (10)$$

where $G$ denotes the number of trajectories , $K$ represents the total number of turns within a trajectory , and $o_k$ signifies the agent's response during the $k$-th turn.

## 5. Experiments

In this section, we conduct comprehensive experiments to evaluate the performance of Strat-Reasoner, focusing on its ability to develop sophisticated strategic reasoning which is generalizable across diverse multi-agent environments.

### 5.1. Experimental setup

**Settings:** We use Qwen3-4B (Yang et al., 2025), a state-of-the-art open-source large language model, as our backbone model for training. Our experimental framework focuses on three distinct two-player strategic environments: (i) perfect-information adversarial games represented by Tic-Tac-Toe, where we evaluate performance against Monte Carlo Tree Search (MCTS). (ii) imperfect-information adversarial games such as KuhnPoker, benchmarked against Nash Equilibrium solutions. (iii) imperfect-information cooperative games like MiniHanabi, which serves to assess coordination capabilities under partial observability. Detailed configurations for these environments are provided in the Appendix D. All evaluation results were computed across more than 500 games.

**Baselines**: To evaluate the effectiveness of our approach, we compare our model against a comprehensive set of baselines. These include larger open-source models (Qwen3-8B, Qwen3-32B, and Gemma3-12B (Team et al., 2025a)), leading closed-source models (GPT-5-mini (Singh et al., 2025) and Gemini-2.5-flash (Comanici et al., 2025)), and specialized high-performance baselines such as SPIRAL (Liu et al., 2025) and MARSHAL (Yuan et al., 2025), both of which are developed based on the self-play paradigm and achieved state-of-the-art performance at the time of their release.

**Evaluation Metrics**: We employ the same return calculation methodology as OpenSpiel (Lanctot et al., 2019), where the specific calculation logic varies across different game environments. This ensures our results are directly comparable with established baselines in the multi-agent reinforcement learning (MARL) community. Detailed information is provided in the Appendix D.

*Table 1.* Average normalized game score comparison. For adversarial games, the two scores corresponds to results of **first-move** and **second-move**. Our Strat-Reasoner is tested under turn orders identical to those encountered during training. For MiniHanabi, a single collective score is utilized to reflect the shared returns common to all participating agents. We categorize models into Closed-source and Open-source groups. **Bold** indicates the best performance within the Open-source category, while underline denotes the best performance across all tested models.

| Model | Tic-Tac-Toe | | KuhnPoker | MiniHanabi |
| --- | --- | --- | --- | --- |
| | *vs. MCTS Bot* | | *vs. NE Bot* | *Co-op* |
| | 100 sims | 1000 sims | | |
| *Closed-source* | | | | |
| GPT-5-mini | 88.84/89.73 | 82.12/87.10 | 77.72/86.56 | 85.99 |
| Gemini-2.5-flash | 86.63/85.52 | 88.32/82.53 | 92.44/87.79 | 79.63 |
| *Open-source* | | | | |
| Qwen3-4B | 65.24/69.56 | 66.78/68.97 | 70.71/70.15 | 58.47 |
| Qwen3-8B | 69.72/69.39 | 68.62/72.03 | 68.26/70.05 | 69.14 |
| Qwen3-32B | 76.53/76.42 | 70.34/76.88 | 74.95/76.06 | 72.86 |
| Gemma3-12B | 71.61/73.58 | 65.81/80.02 | 66.71/65.01 | 60.25 |
| SPIRAL-4B | 70.45/69.77 | 74.20/67.84 | 58.01/69.38 | 62.65 |
| MARSHAL-4B | 73.47/81.55 | 76.10/**81.22** | 75.05/73.94 | 77.14 |
| **Strat-Reasoner-4B (Ours)** | **90.77**/81.84 | **77.60**/73.12 | **94.04**/**90.47** | **80.19** |

## 5.2. Performance Comparisons with Baselines

To ensure a comprehensive assessment of game-playing performance, we benchmark different models against a standardized suite of formidable, static opponents. All opponents' details can be found in Appendix D. As illustrated in Table 1, which reports the normalized average score per game as our primary evaluation metric, our framework demonstrates remarkable efficacy.

Agents trained under our framework, whether employing first-mover or second-mover strategies, consistently outperform same-series open-source models, as well as models with significantly larger parameter scales. Furthermore, our approach yields substantial improvements over specialized baselines such as SPIRAL and MARSHAL. Impressively, it achieves performance on par with state-of-the-art models including GPT-5-mini and Gemini-2.5-Flash.

Experimental results reveal a clear correlation between agent performance and the turn order used during training. Specifically, this proficiency is most pronounced when the testing sequence aligns with the training configuration. This performance asymmetry underscores the efficacy of our joint training framework, which enables agents to evolve highly specialized strategic capabilities for specific acting sequences. Furthermore, the agents maintain competitive performance in cross-order test scenarios, further validating the framework's generalization robustness across different player positions.

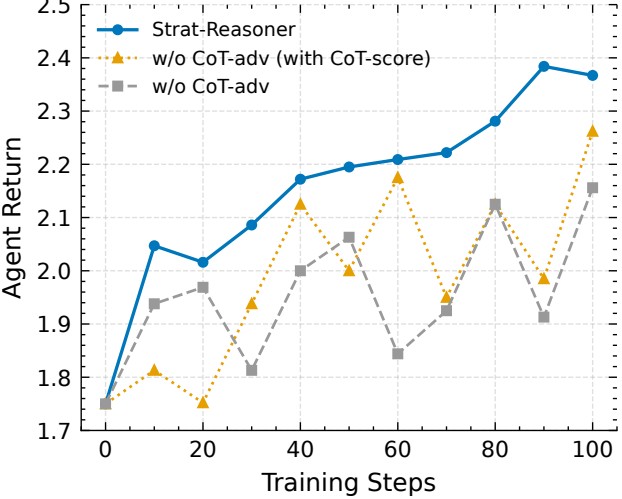

*Figure 4.* Ablation study of MiniHanabi. The curves represent the full Strat-Reasoner (blue), and the variant using raw CoT scores without normalization (yellow), the variant without CoT training signals (grey).

## 5.3. Ablation Studies

To evaluate the design of our Strat-Reasoner framework, we conducted two ablation studies: (1) training exclusively with the recursive prompt module while excluding training signals from CoT. (2) utilizing the recursive prompt module while incorporating raw CoT scores directly as the advantage in the training signal without normalization.

Figure 4 illustrates the performance curves in the MiniHan-

*Table 2.* Average normalized game score comparison across OOD game environments. In adversarial scenarios, scores are recorded separately for the first-mover and second-mover. To ensure consistency, the evaluation of Strat-Reasoner maintains the same turn-order dynamics as the training phase. For SimpleHanabi, we also report a single collective score. **Bold** indicates the best performance within the Open-source category, while underline denotes the best performance across all tested models.

| Models | ConnectFour | | LeducHoldem | SimpleHanabi |
| | *vs. MCTS Bot* | | *vs. NE Bot* | *Co-op* |
| | 10 sims | 100 sims | | |
| --- | --- | --- | --- | --- |
| *Closed-source* | | | | |
| GPT-5-mini | 99.93/97.95 | 96.97/72.10 | 80.60/95.08 | 72.77 |
| Gemini-2.5-flash | 78.74/82.70 | 64.90/66.14 | 83.51/96.96 | 73.10 |
| *Open-source* | | | | |
| Qwen3-4B | 63.20/57.01 | 62.77/58.39 | 61.35/56.18 | 55.53 |
| Qwen3-8B | 64.10/62.34 | 58.00/64.88 | 60.93/64.41 | 64.32 |
| Qwen3-32B | 56.62/63.63 | 58.11/58.12 | 67.60/67.51 | 65.35 |
| Gemma3-12B | 50.50/57.01 | 61.14/58.66 | 60.55/60.13 | 55.71 |
| SPIRAL-4B | 68.00/60.85 | 68.71/59.24 | 56.21/55.85 | 55.55 |
| MARSHAL-4B | 66.98/69.94 | 70.67/67.00 | 66.42/**69.22** | 65.65 |
| **Strat-Reasoner-4B (Ours)** | **70.39/70.26** | **75.93/69.26** | **70.12**/66.64 | **68.63** |

abi environment, demonstrating that the framework's superiority stems primarily from the integration of CoT training signals. The full Strat-Reasoner framework exhibits superior strategic capability and a consistent upward trajectory, significantly surpassing all ablated baselines. Specifically, excluding CoT signals restricts the agent's strategic development, preventing it from exceeding baseline levels. Moreover, the use of unnormalized CoT scores induces substantial training instability and performance degradation. These findings validate the indispensability of the full framework, confirming that the synergy between structured prompts and CoT-based advantage normalization is fundamental to enhancing both strategic depth and training robustness.

### 5.4. Generalization Studies

To evaluate the generalization capabilities of the Strat-Reasoner framework, we conduct testing on game environments that were unseen during the training phase. Specifically, agents trained on Tic-Tac-Toe are evaluated on the more challenging Connect Four, those trained on Kuhn-Poker are tested on LeducHoldem, and for agents trained on MiniHanabi, we employ the more complex SimpleHanabi for generalization assessment. We also benchmark our framework against the baseline models under the same conditions, detailed configurations for these generalization environments are provided in the Appendix D.

As illustrated in Table 2, our framework exhibits robust generalization capabilities when transferred to out-of-distribution (OOD) environments. Despite the significantly increased difficulty of the test scenarios, our agent consistently outperforms baseline models with comparable parameter counts and even surpasses substantially larger models in certain instances. This performance stability indicates that our framework does not overfit to specific training settings; rather, it effectively enhances strategic reasoning across diverse game environments.

In particular, the consistent gains across ConnectFour, LeducHoldem, and SimpleHanabi suggest that the learned recursive reasoning pattern can transfer to settings with larger action spaces, hidden information, or cooperative coordination. Consequently, our agent maintains high-level strategic thinking and reasoning capabilities even in unseen and more challenging contexts.

### 5.5. Computational Cost

All experiments were conducted on a workstation equipped with dual NVIDIA A800 (80GB) GPUs.

Although we employ Low-Rank Adaptation (LoRA) to significantly reduce the number of trainable parameters (see Appendix D for details), the training process is still constrained by the multi-turn nature of trajectories and the extensive token length per output.

Consequently, the primary computational bottleneck shifts to the rollout phase, which accounts for 50%–60% of the total time per global step. Nevertheless, our framework remains efficient, requiring 0.3 to 0.7 GPU hours per global step. In the most complex scenarios, such as MiniHanabi, the entire pipeline typically spans 100 wall-clock hours. Our subsequent research may explore asynchronous rollout protocols and trajectory pruning to optimize the reasoning process and further reduce total wall-clock time.

## 6. Conclusion

In this work, we present Strat-Reasoner, an RL framework that incentivizes LLMs' strategic reasoning capabilities in multi-agent games. We integrate recursive reasoning into the strategic decision-making process and establish a novel paradigm that utilizes and optimizes reasoning processes in a fine-grained, reward-model-free manner. Strat-Reasoner achieves state-of-the-art performance by consistently surpassing representative open-source baselines and rivaling premier closed-source models across diverse game environments. These results validate the framework's effectiveness in fostering robust strategic reasoning in LLMs.

While the results of two-agent settings have already significantly demonstrated the superiority of Strat-Reasoner, scenarios involving more agents are also prevalent in real-world problems. Our method is inherently extensible to $N$-agent settings by expanding the recursive reasoning structure; however, as this requires further engineering adaptation, we leave it as future work.

## Acknowledgements

This work was sponsored by the Guangdong Basic and Applied Basic Research Foundation (No. 2025A1515010247), and the CCF-Kuaishou Large Model Explorer Fund (NO. CCF-KuaiShou 2025004), and the National Natural Science Foundation of China (No. 62506133).

## Impact Statement

Our work aims to enhance the strategic reasoning capabilities of Large Language Models in complex game-theoretic scenarios, potentially contributing to advances in fields such as scientific research and artificial intelligence. However, such advancements are inherently dual-use. Models with superior strategic proficiency could be exploited irresponsibly, such as facilitating illicit transactions or other harmful applications. We urge researchers and practitioners to implement safeguards when deploying our methods. Overall, this paper advances fundamental techniques in machine learning; its societal impact will depend on responsible deployment and continued ethical considerations by the community.

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

## A. Use of LLMs

Large Language Models (LLMs) were employed solely for proofreading, grammatical correction, and linguistic refinement to enhance the clarity of this paper. The core intellectual content, data analysis, and conclusions were developed independently by the authors without AI involvement. The authors remain fully accountable for the integrity of the work.

## B. Future work

Our framework is tailored for LLM-based multi-agent systems. Future research will extend Structured Chain-of-Thought to environments involving more than two agents and propose a universal recursive thinking prompting paradigm. Furthermore, we aim to achieve breakthroughs in both asynchronous and synchronous action scenarios, striving to develop a genuine LLM-based Multi-Agent Reinforcement Learning (MARL) algorithm.

## C. Scope, Environment setting, and Extensibility

**Scope.** Our formal problem setting is Two-player Alternating Markov Games, as defined in Section 3.1. This setting is a fundamental and widely used multi-agent formulation: although only one agent acts at each turn, the value of each decision depends on the evolving policy, hidden intent, and future response of the other agent. Therefore, the setting already exposes the core difficulties targeted by Strat-Reasoner, including opponent non-stationarity, the absence of direct supervision for the reasoning process, and credit assignment across multiple turns of interdependent reasoning. We consequently state our empirical claims within this scope, while treating broader $N$-agent validation as future work.

**Environment setting.** The three training environments are chosen to cover complementary game-theoretic regimes rather than variations of a single game family. Tic-Tac-Toe is a perfect-information competitive game, KuhnPoker is an imperfect-information competitive game, and MiniHanabi is an imperfect-information cooperative game. This combination tests whether the proposed recursive reasoning signal remains useful when the agent must reason about visible tactical threats, hidden private information, and cooperative intent under partial observability. In addition, we evaluate out-of-distribution transfer on ConnectFour, LeducHoldem, and SimpleHanabi. These OOD environments preserve the high-level strategic regime of the corresponding training games while increasing the state space, action space, or information complexity. The resulting evaluation therefore probes whether Strat-Reasoner learns reusable strategic reasoning patterns instead of only memorizing environment-specific policies.

**Extension to $N$-agent settings.** A naive extension of recursive reasoning to $N$ agents would require modeling all pairwise belief relations, which can introduce unnecessary computational cost and high-variance comparison targets. Strat-Reasoner can instead be extended through a targeted reasoning mechanism. Concretely, the structured reasoning block can be augmented with a target identifier, so that the ego agent explicitly specifies which opponent or teammate is being modeled in its `OpponentIntent` and `OpponentPrediction` fields. In alternating games, this target may be selected from agents who acted in recent turns; in simultaneous-move games, the target can be selected from the agents whose previous actions are revealed at the current update. The centralized CoT comparison module can then retrieve the corresponding target agent's `MyIntent` and `MyPrediction` as ground truth for the specified semantic pairs.

Under this targeted formulation, the core mechanism of Strat-Reasoner remains unchanged: the model still produces structured recursive beliefs, the training process still compares predicted beliefs against realized reasoning or actions, and the policy is still optimized with the hybrid advantage. If multiple targets are selected, their alignment scores can be averaged to form a turn-level CoT score. This avoids exhaustive all-to-all comparison while preserving the essential training signal. A full empirical study of such $N$-agent settings requires standardized benchmarks and comparable LLM-based MARL baselines, so we leave it as an important direction for future work.

## D. Experiment details

**Training Setup.** The implementation of the Strat-Reasoner framework is primarily built upon ROLL (Wang et al., 2025), a robust open-source codebase for Reinforcement Learning from LLM Feedback. Leveraging ROLL's native support for agentic, multi-turn rollouts, we established a solid foundation for our system. Furthermore, all game environments are integrated via OpenSpiel (Lanctot et al., 2019) to guarantee the correctness and standardization of the underlying game logic. All prompts related to game information are consistent with MARSHAL (Yuan et al., 2025).

**Framework.** We perform reinforcement learning fine-tuning (RLFT) on the Qwen3-4B model using Low-Rank Adaptation (LoRA) (Hu et al., 2022). Leveraging the native support for vLLM (Kwon et al., 2023)—a high-speed inference engine—within the ROLL framework, we employ vLLM's multi-LoRA capability as our core infrastructure. This architecture facilitates the seamless hot-swapping of LoRA adapters, enabling the simultaneous training of two LLM agents. All experiments are conducted on a single machine equipped with dual NVIDIA A800 GPUs.

**On-policy role-specific training.** Strat-Reasoner uses a strict on-policy training protocol. For each two-player environment, we instantiate two distinct LLM agents corresponding to the two player roles. During rollout, these agents interact with each other in the OpenSpiel environment and generate trajectories under the current policies. After trajectory collection, each agent is fine-tuned only on its own role-specific trajectory data. This independent training regime is different from shared-parameter self-play: it preserves the non-stationarity induced by another learning agent and better matches heterogeneous multi-agent scenarios, where different agents may have distinct reasoning processes and policies.

**Evaluation protocol.** All results reported in Tables 1 and 2 are obtained by dynamically running the trained models in the corresponding OpenSpiel game environments. They are not computed from static datasets or offline action labels. For adversarial games, each trained role is evaluated under the same turn order used during training against the specified static opponent, such as MCTS bots for Tic-Tac-Toe and ConnectFour or NE/CFR-based bots for poker environments. For cooperative Hanabi variants, the reported score is the collective game return. Each evaluation entry is averaged over more than 500 games. The normalized scores in the main tables are computed from the raw OpenSpiel returns so that different environments can be compared in a unified scale.

**Stopping criterion.** Following common practice in LLM-based MARL training, we use online validation to select the final model. Every 10 training steps, the current policy is evaluated against a fixed validation opponent, e.g., an MCTS-100 simulations bot for Tic-Tac-Toe. Training is stopped when the validation performance converges or starts to drop, which prevents overfitting to the recent on-policy trajectories. The final results in Tables 1 and 2 are produced by the last checkpoint saved at the completion of training under this early-stopping criterion.

**Reward Design.**

**Intrinsic Game Rewards.** We utilize the default game rewards provided by the OpenSpiel framework as our primary reward signals. Specifically, in Tic-Tac-Toe, the rewards for a win, draw, and loss are $+1$, $0$, and $-1$, respectively. In Kuhn Poker, the payoff ranges from $-2$ to $+2$ depending on the specific game state. For MiniHanabi, agents receive a shared reward of $+1$ for each successfully played card, reaching a maximum total reward of $4$.

**Format Regularization.** Following the methodology of DeepSeek-R1 (Guo et al., 2025), we enforce structural constraints on the model's output by granting a nominal reward $(+0.05)$ for correct formatting and a severe penalty $(-10.0)$ for invalid actions, the latter of which also triggers an episode termination.

**Length Penalty.** We employ a turn-level verbosity penalty to promote conciseness similar to Kimi k1.5 (Team et al., 2025b). This penalty scales linearly for any response exceeding a predefined length threshold. The penalty is calculated as:

$$r_{\text{length}}(l) = \alpha \cdot \min\left(0, 1 - \frac{l - l_{\min}}{l_{\max} - l_{\min}}\right) \tag{11}$$

where we set $l_{\min} = 11$, $l_{\max} = 2048$, and the scaling coefficient $\alpha = 0.5$.

**Environment Agent Details.**

**Tic-Tac-Toe and ConnectFour.** MCTS agents with varying simulation counts (100/1000 for Tic-Tac-Toe, 10/100 for Connect Four) to test against different strengths.

**KuhnPoker.** The exact Nash Equilibrium (NE) policy.

**LeducHoldem.** A NE policy approximated by $5 \times 10^8$ iterations of Counterfactual Regret Minimization (CFR).

# E. Prompt

---

**Listing E.1: Structured Reasoning Prompt**

STATE SUMMARY TASK:
In your `<think>` section, You MUST analyze the current situation, then provide a concise turn summary using the exact format below, and output in EXACT format. [`state_summary`: ...] (1 sentence, no tables). This summary is required every turn.

RECURSIVE REASONING TASK:
In your `<think>` section, You MUST also complete the following reasoning template. Fill in the content inside ALL brackets [] with your analysis:

1. Because {your reasoning}, I believe the opponent's intent is [`OpponentIntent`: ].

2. Because {your reasoning}, I believe the opponent predicts my action will be [`OpponentPrediction`: ].

3. Because {your reasoning}, My current intent is [`MyIntent`: ], my chosen action is [`MyAction`: ], and I predict the opponent's next action will be [`MyPrediction`: ].

RESPONSE INSTRUCTIONS:
You MUST follow this EXACT output structure:

1. End your thinking process with `</think>` tag.

2. Write your reasoning process inside the think tags.

3. Complete ALL bracketed fields [`field:your_content`] in your thinking process.

*Example:*
`<think>your thinking here ...</think><answer>your answer here</answer>`

STRICT RULES:

- Choose the best action based on the game state and your thinking.

- No self-correction loops; do not revisit earlier sentences.

- Keep your thinking process CONCISE and EFFECTIVE.

- Response that do not follow this exact format, or overlong response, will result in IMMEDIATE LOSS of the game.

- NO text after `</answer>`.

---

**Listing E.2: CoT Scoring Prompt for TicTacToe**

**system prompt:**
You are an AI agent specialized in semantic analysis and behavioral intent recognition. Your expertise lies in evaluating the accuracy of predictions about opponent intentions in strategic games.

---

**user prompt:**
EVALUATION CONTEXT:

1. In a Tic-Tac-Toe game, two players take turns making moves while reasoning about each other's intentions.

2. Each player generates predictions about the opponent's intent (what they plan to do) and future actions.

3. Your task is to evaluate how accurately one player's prediction matches the opponent's actual stated intent.

SCORING CRITERIA:

- **0.0-0.3:** Prediction is completely inconsistent with reality (wrong direction or unrelated)

- **0.3-0.6:** Prediction is partially correct (captures some aspects but misses key points)

- **0.6-0.8:** Prediction is mostly accurate (main intent captured with minor differences)

- **0.8-1.0:** Prediction is highly consistent or exactly matches reality

STRICT RULES:

- Your score MUST be a decimal between 0.0 and 1.0

RESPONSE INSTRUCTIONS:
You MUST follow this EXACT output structure:

1. Directly Output your final score as `<answer>`YOUR_SCORE`</answer>`

2. Your response MUST end with `</answer>` - this is MANDATORY

Example output: `<answer>`YOUR_SCORE`</answer>`

---

**Listing E.3: CoT Scoring Prompt for KuhnPoker**

**system prompt:**
You are an AI agent specialized in semantic analysis and behavioral intent recognition. Your expertise lies in evaluating the accuracy of predictions about opponent intentions in strategic games.

---

**user prompt:**
EVALUATION CONTEXT:

1. In Kuhn Poker, two players each receive one hidden card (J/Q/K) and play a single round of betting.

2. Each player generates predictions about the other player's intent (e.g., bluffing vs value, likely bet/call/fold) and future betting actions based on observed moves.

3. Your task is to evaluate how accurately one player's prediction matches the other player's actual stated intent in the same situation.

SCORING CRITERIA:

- **0.0-0.3:** Prediction is completely inconsistent with reality (wrong direction or unrelated)

- **0.3-0.6:** Prediction is partially correct (captures some aspects but misses key points)

- **0.6-0.8:** Prediction is mostly accurate (main intent captured with minor differences)

- **0.8-1.0:** Prediction is highly consistent or exactly matches reality

STRICT RULES:

- Your score MUST be a decimal between 0.0 and 1.0

RESPONSE INSTRUCTIONS:
You MUST follow this EXACT output structure:

1. Directly Output your final score as `<answer>`YOUR_SCORE`</answer>`

2. Your response MUST end with `</answer>` - this is MANDATORY

Example output: `<answer>`YOUR_SCORE`</answer>`

**Listing E.4: CoT Scoring Prompt for Hanabi**

**system prompt:**
You are an AI agent specialized in semantic analysis and behavioral intent recognition. Your expertise lies in evaluating the accuracy of predictions about opponent intentions in strategic games.

**user prompt:**
EVALUATION CONTEXT:

1. In Hanabi, players cooperate to build fireworks by playing cards in order, but cannot see their own hands and must infer them from hints.

2. Each player generates predictions about the other player's intent (e.g., why they gave a hint, what card they plan to play/discard, what they believe about hidden cards) and future cooperative actions.

3. Your task is to evaluate how accurately one player's prediction matches the other player's actual stated intent given the shared game context.

SCORING CRITERIA:

- **0.0-0.3:** Prediction is completely inconsistent with reality (wrong direction or unrelated)

- **0.3-0.6:** Prediction is partially correct (captures some aspects but misses key points)

- **0.6-0.8:** Prediction is mostly accurate (main intent captured with minor differences)

- **0.8-1.0:** Prediction is highly consistent or exactly matches reality

STRICT RULES:

- Your score MUST be a decimal between 0.0 and 1.0

RESPONSE INSTRUCTIONS:
You MUST follow this EXACT output structure:

1. Directly Output your final score as `<answer>YOUR_SCORE</answer>`

2. Your response MUST end with `</answer>` - this is MANDATORY

Example output: `<answer>YOUR_SCORE</answer>`

# F. Hyperparameters

Hyperparameter settings are shown in Table 3, which may vary slightly depending on the specific environment.

*Table 3.* Hyperparameters

|  | Parameter | Value |
|---|---|---|
| **Training Settings** | Dtype | bf16 |
|  | Train Batch Size | 128 |
|  | Optimizer | Adam |
|  | Adam parameters $(\beta_1, \beta_2)$ | (0.9, 0.95) |
|  | Learning Scheduler | Cosine |
|  | Learning Rate | $5 \times 10^{-6}$ |
|  | Weight Decay | 0.05 |
|  | Gradient Norm Clip | 1.0 |
|  | Lora Rank | 16 |
|  | Lora Alpha | 32 |
|  | Lora Dropout | 0.05 |
|  | Micro Rollout Size | 2 |
|  | CoT Advantage Weight | 0.2 |
| **RL Settings** | Sampling Temperature | 0.5 |
|  | PPO Epochs | 1 |
|  | (top P, top k) | (0.9, 100) |
|  | KL Loss | true |
|  | KL Loss Coefficient | 0.15 |
|  | Dual Clip Loss | true |
|  | PPO Policy Clip | 0.2 |
|  | Lambda | 0.95 |
|  | Gamma | 1 |

# G. Hyperparameter Sensitivity and Evaluator Robustness

**CoT advantage weight.** The CoT advantage weight $\omega$ controls the balance between the sparse but unbiased return-based signal and the dense but local CoT-based signal. We select $\omega = 0.2$ based on empirical sensitivity analysis in MiniHanabi, summarized in Table 4. When $\omega$ is too small, the CoT signal is not strong enough to consistently guide recursive reasoning. When $\omega$ is too large, the policy can over-emphasize local alignment with the opponent's immediate reasoning, which may hurt the long-term game objective. The setting $\omega = 0.2$ provides the best final performance among the tested values and is therefore used in the main experiments.

*Table 4.* Sensitivity analysis of the CoT advantage weight $\omega$ in MiniHanabi. Returns are reported from the training curves.

| $\omega$ | Best return | Final return | Observation |
|---|---|---|---|
| 0.1 | 2.188 | 2.150 | CoT guidance is useful but relatively weak, yielding slower and less stable improvement. |
| 0.2 | **2.384** | **2.367** | Best trade-off between local recursive-reasoning supervision and the final game objective. |
| 0.3 | 2.320 | 1.812 | Strong local CoT pressure can degrade final performance after transient gains. |

**Micro-rollout size.** Micro-rollouts provide the group of candidate reasoning paths used to normalize the immediate CoT score and estimate the CoT advantage. Increasing the micro-rollout size can reduce the variance of this estimate, similar to the role of group samples in GRPO, but it also increases the number of parallel generations and CoT-comparison calls required at each update. Figure 5 summarizes this empirical performance-cost trade-off. Moving from a very small group to $M = 2$ improves the quality of the relative CoT baseline and yields a clear performance gain with moderate additional cost. Beyond this point, the marginal performance improvement becomes small while the computation grows rapidly, because the rollout and scoring pipeline starts to saturate under a large number of concurrent requests. Thus, $M = 2$ lies near the knee point of the Pareto frontier: it keeps most of the benefit of variance reduction while avoiding the high overhead of larger micro-rollout groups. We therefore use $M = 2$ throughout the main experiments.

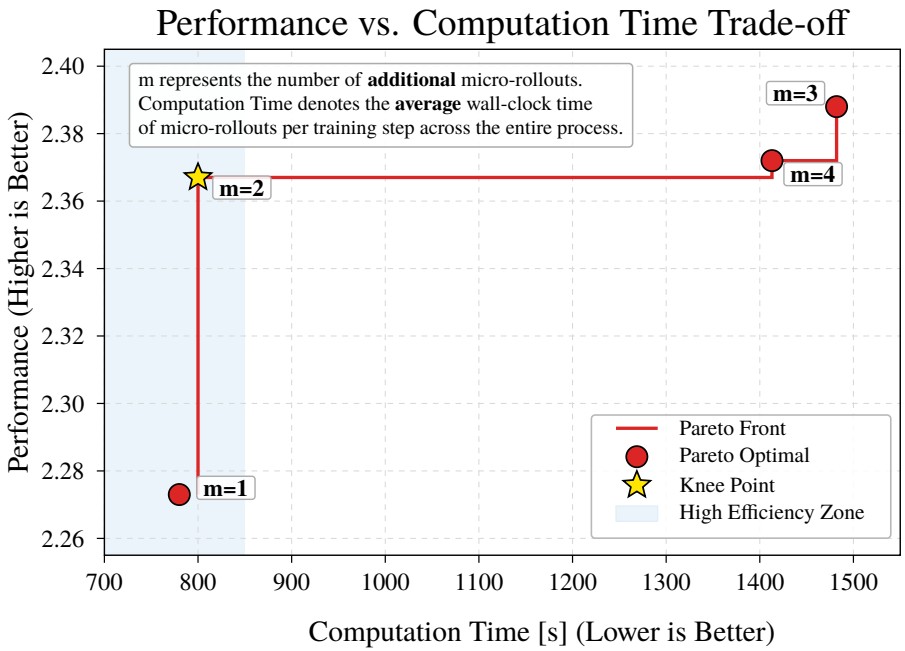

*Figure 5.* Performance-cost Pareto trade-off for the micro-rollout size. Larger micro-rollout groups can provide lower-variance CoT advantage estimates, but the additional parallel generation and scoring cost grows quickly after the knee point.

**Evaluator robustness.** The centralized CoT comparison module uses the base model as an evaluator for efficiency. This does not require the evaluator to solve the strategic decision problem again: the comparison is decomposed into independent semantic pairs, such as matching an `OpponentIntent` prediction against the target agent's realized `MyIntent`. The evaluator is therefore asked to score semantic consistency under the game rules, rather than to provide an open-ended preference judgment over complete trajectories. Moreover, the evaluator context is separated from the policy-generation context, reducing the risk of direct self-alignment between action generation and scoring.

To check that this scoring signal is not overly dependent on the base evaluator, we additionally score representative CoT comparison pairs with several stronger or different models. Table 5 shows one example from Tic-Tac-Toe. The scores are highly consistent across evaluators, suggesting that the semantic-pair comparison is robust for the type of localized reasoning alignment used by Strat-Reasoner.

*Table 5.* Evaluator robustness example for the field `OpponentIntent`. Prediction: "Prevent opponent's diagonal." Ground truth: "placing X in (0, 2) to block diagonal."

| Evaluator | Qwen3-4B | Qwen3-32B | DeepSeek-v3.1 | GPT-5 | Claude-4.5 | Gemini-3.1 |
|---|---|---|---|---|---|---|
| Score | 0.90 | 1.00 | 0.85 | 0.95 | 0.90 | 0.90 |

# H. Case study

> **Tic-Tac-Toe Strategic Reasoning**
>
> **GAME STATE:**
>
> 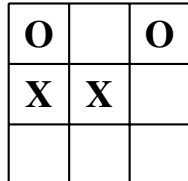
>
> - **Opponent Move:** $O(0, 2)$
> - **Current Turn:** Player X
>
> **REASONING TRACE:**
>
> Okay, let me try to figure out what move I should make now. So, the current game state is:
>
> ```
> O _ O
> X X _
> _ _ _
> ```
>
> Looking at the board, the rows are:
>
> ```
> Row 0: O _ O
> Row 1: X X _
> Row 2: _ _ _
> ```
>
> So, the empty cells are (0,1), (1,2), (2,0), (2,1), (2,2).
>
> The possible moves for X are those positions. Let me check if there's any immediate winning opportunity.
>
> First, checking for lines. For X, they have two in a row in row 1 (positions 1,0 and 1,1). If X can get the third in row 1, column 2 (position 1,2), that would complete a horizontal line. But wait, position 1,2 is currently empty. However, the opponent might block that. Wait, but in the previous move, the opponent placed O at (0,2). So maybe they're trying to prevent X from completing a line.
>
> Alternatively, looking for diagonals. X has a potential diagonal from (0,0) to (1,1) to (2,2), but (0,0) is O, so that doesn't help. Another diagonal could be (0,2) which is O, but that's already taken. Maybe not helpful.
>
> What about columns? Column 0 has O, X, _. Not helpful. Column 1 has _, X, _. Not helpful. Column 2 has O, _, _. Hmm.
>
> **Wait, perhaps the key is to block the opponent's possible threats. The opponent just placed O at (0,2). Looking back, the previous move was O putting at (0,2). Now, the opponent might be trying to create a line. Let me see: the opponent's last move was (0,2), making the first row O _ O. That gives them two corners.**
>
> They might be aiming for a diagonal. If they put O at (2,0), that would form a diagonal from (0,2) to (1,1) to (2,0)? Wait, no. Wait, (0,2) is O, and if they put O at (2,0), that's a diagonal. But that's only two O's. Alternatively, maybe they want to complete a row or column.
>
> But since it's my turn, I need to decide where to place X. Let's look again. Possible moves are (0,1), (1,2), (2,0), (2,1), (2,2).
>
> If I play at (1,2), that completes the second row's third cell. Since row 1 is XX_, adding X at (1,2) would make XXX, which is a win. But is that safe? Well, the opponent hasn't blocked that yet. Wait, the opponent's last move was O at (0,2).

```
Are they going to block that?

 Alternatively, if I play at (0,1), that would fill the middle of the first row.
Then the first row becomes O X O. That's not a win, but maybe setting up for future
 moves.

 Or playing at (2,0): that's the lower left corner. Could that lead to a diagonal?
If I place X at (2,0), then if the opponent blocks elsewhere, maybe I can connect
with other Xs.
```

**Another thought:  the opponent may be trying to create a threat.  After their move at (0,2), they have two O's in the first row.  If I don't block, they might finish that row by placing O at (0,1), but that's not possible because the opponent's next move is their**
**turn.** Wait, no, this is my turn. The opponent just moved, and now it's my turn. So the opponent's move was at (0,2), and now I'm choosing where to place X.

```
 Let me consider possible scenarios. Suppose I go for (1,2) to complete the row.
That seems like a strong move because it directly leads to a win. Unless the
opponent can block. But the opponent's next move would be after mine. Wait, no, the
 opponent is the one who plays O. So after I place X at (1,2), the opponent gets to
 respond. But if I place X there, that's a direct win unless they can stop me. But
since the opponent isn't able to block immediately, maybe that's the right choice.

 Alternatively, if I place X at (2,2), that's the bottom right corner. It could be
part of a diagonal. But the opponent might block that later.

 Hmm. Which option is better? Completing the row (1,2) is more straightforward. Let
me confirm: row 1 is X X _, so adding X at (1,2) would give three in a row. Yes.
Therefore, that's a winning move. But why didn't the opponent block that? Because
they haven't had a chance yet. So, I should go for that.
```

**Therefore, the correct move is X(1,2).**

**[state_summary:  X has two in a row on the middle row, opponent has two in the top row corners.]**

**1.  Because [the opponent has secured two corners in the top row], I believe the opponent's intent is [OpponentIntent:  To complete a row or diagonal].**

**2.  Because [the opponent likely anticipates blocking my potential row completion], I believe the opponent predicts my action will be [OpponentPrediction:  Placing X in (1,2) to finish the middle row].**

**3.  Because [I prioritize securing a quick win while countering threats], My current intent is [MyIntent:  Finish the middle row], my chosen action is [MyAction: X(1,2)], and I predict the opponent's next action will be [MyPrediction:  Blocking by placing O in (1,2)].**

