# OpenReview forum: "Strat-Reasoner: Reinforcing Strategic Reasoning of LLMs in Multi-Agent Games"
_ICML.cc/2026/Conference — ICML 2026 regular_

### Official Review · Reviewer_mKFL · 2026-02-23

**Soundness:** 3
**Presentation:** 4
**Significance:** 3
**Originality:** 3
**Overall Recommendation:** 5
**Confidence:** 4

**Summary:**

This paper proposes Strat-Reasoner, a reinforcement learning framework designed to enhance strategic reasoning abilities of LLMs in multi-agent games. The authors aim to address a crucial open question in LLM-based multi-agent reinforcement learning: how to provide effective intermediate training signals for long-horizon strategic reasoning under opponent non-stationarity? This work introduces a central concept from game-theoretic intelligence, recursive reasoning, to model the beliefs of the other agents. Specifically, this involves explicitly structuring the ego agent’s thought process into opponent intent prediction, higher-order belief prediction (predicting the opponent's belief of itself), and future action prediction. To stabilize training, the framework introduces a centralized CoT comparison mechanism that rewards the ego agent by evaluating the alignment between the ego agent’s predicted beliefs and the opponent’s actual reasoning, and integrates these signals with return-based rewards through a hybrid advantage estimator. Experiments across competitive and cooperative games demonstrate consistent performance gains over strong open-source baselines and competitive generalization to unseen environments, suggesting that explicitly optimizing structured strategic reasoning can substantially improve LLM performance in multi-agent settings.

**Compliance With Llm Reviewing Policy:**

Affirmed.

**Final Justification:**

Thank you for the detailed and thoughtful response. I appreciate the clarifications and additional explanations provided. They addressed my main concerns well, and I am happy to maintain my current score.

**Key Questions For Authors:**

1. What is the impact of the choice of the evaluator model on the performance of the trained model?
2. Is Strat-Reasoner trained separately for each player role? If so, is it trained against LLM agents or fixed bots? Why adopt such a training regime?
3. What is the impact of the group size in the micro rollout?

**Limitations:**

1. Dependency on the evaluator model.
2. Non-trivial computational overhead during rollout.
3. Limited validation beyond game-based settings.

**Strengths And Weaknesses:**

This is a technically solid piece of work, and I want to start by highlighting what I think is very good.

Strengths:

1. This paper is clear and well-motivated. The authors clearly identify the key challenges of extending RL of LLM to multi-agent strategic tasks, including the non-stationarity introduced by other agents, the sparse and delayed rewards, and designing fine-grained credit assignment that incentivizes strategic reasoning.
2. The paper proposes a structured approach to introducing recursive reasoning into RL for strategic LLMs, inspired by cognitive hierarchy theory. While recursive reasoning in LLMs has been previously introduced, successfully performing training under such a paradigm is unprecedented (to the best of my knowledge). Explicitly decomposing reasoning into "Opponent Intent", "Opponent Prediction", "My Intent", and "My Prediction" is conceptually clean and aligns well with alternating Markov dynamics in turn-based multi-player games.
3. The centralized CoT comparison module and hybrid advantage estimation are well-designed. The CoT comparison module offers turn-level reasoning supervision, addressing the sparsity of outcome-based rewards. The hybrid advantage estimation successfully integrates this turn-level reasoning reward with the sparse outcome reward, grounding the recursive reasoning in the final objective of the games.
4. The paper demonstrates consistent improvements over strong open-source baselines across competitive and cooperative games and across diverse model backbones and sizes. The results also demonstrate generalization to unseen games, indicating generalizable reasoning skills.
5. The paper is very well-presented. The writing is intuitive and very easy to follow. The authors also perform a very good job of visually demonstrating recursive reasoning and the proposed approach.

With that said, I believe some questions should be further studied and discussed.

Weaknesses:

1. The CoT comparison module directly uses the base model as the evaluator. However, different evaluator models may exhibit different judgments, which could affect the resulting CoT scores and, in turn, the trained model's strategic reasoning capability. Using the same backbone for both policy and evaluation may also introduce bias or unintended self-alignment effects. It would be beneficial to ablate different evaluator models to assess the robustness of the design of the training signal.
2. Potential sensitivity of evaluation performance to turn order in training. For start, I would like to point out that the results presented in the main text (Table 1 and Table 2) do not reflect the different performance of training Strat-Reasoner as player 0 and player 1. Therefore, the conclusion "experimental results reveal a clear correlation between agent performance and the turn order used during training" from line 357 is confusing. It's only from the raw results in the appendix that I realize Strat-Reasoner is actually trained separately for each player role, different from self-play baselines (MARSHAL and SPIRAL) that use a single model for all player roles. If my understanding is correct and Strat-Reasoner does in fact exhibit a sensitivity to the turn order in training, then a critical question is whether Strat-Reasoner demonstrates the same advantage over baseline methods when using the same self-play setting? If not, what is the cause?
3. Computational overhead. Strat-Reasoner adopts a micro rollout strategy that performs a group of one-step rollouts in each turn to estimate a baseline for normalizing the intermediate reasoning rewards. The paper would benefit from a further analysis on how this group size affects the computational cost and the resulting performance.
4. The proposed framework is evaluated primarily on structured two-player games. While useful for controlled experimentation, this remains relatively constrained and well-defined environments. It is unclear how well the method would transfer to more real-world agentic tasks or multi-agent systems.

---

> ### Author Rebuttal · Authors · 2026-03-29
>
> Hi Reviewer mKFL,
>
> We are highly encouraged by your positive evaluation and insightful feedback!
>
> ---
>
> > ## W1 & Q1: Concerns about the evaluator model.
>
> > **Action**: We will add the analysis of the evaluator model in Appendix.
>
> In our method, the CoT comparison is explicitly disentangled into **independent semantic pairs**. As the agent, the base model performs recursive reasoning, while as the evaluator, it only evaluates semantic similarity between pairs, which is **independent of strategic reasoning and largely differs from open-ended subjective scoring**. Furthermore, the **contexts of the two tasks do not overlap**. Thus using the base model as the evaluator **does not introduce systematic bias or self-alignment effects**.
>
> Nevertheless, we fully agree that empirically validating the robustness of the training signal under different evaluator models is valuable. **We benchmarked the scoring performance across various models**. An example is shown below:
>
> Evaluated field: opponent_intent
>
> Prediction: Prevent opponent’s diagonal
>
> Ground Truth: placing X in (0, 2) to block diagonal
>
> | model | Qwen3-4B | Qwen3-32B | Deepseekv3.1 | GPT-5 | Claude4.5 | Gemini3.1 |
> |--|--|--|--|--|--|--|
> | score | 0.85 | 1.0 | 0.85 | 0.95 | 0.9 | 0.9 |
>
> As demonstrated, other models yield highly consistent scores with Qwen3-4B. Consequently, we opted for the base model to optimize computational resources and deployment efficiency.
>
> ---
>
> > ## W2 & Q2: Self-play vs Role-Specific training.
>
> > **Action**: We will further clarify the training details in Appendix C, and treat the self-play setup as an optional baseline.
>
> We appreciate your careful examination of our experimental design. You are correct that Strat-Reasoner is trained separately for each player role. Specifically, **two distinct LLM agents interacted with each other** to generate trajectories and then **fine-tuned respectively on their own trajectory data**.
>
> While self-play methods improve training efficiency, they often mask the non-stationarity caused by unknown opponents in MARL. However, real-world multi-agent scenarios typically involve heterogeneous agents with distinct, independent reasoning processes. As a result, **we adopted the independent training regime to reflect real-world multi-agent dynamics, demonstrating its broader applicability**.
>
> Regarding your question about evaluating Strat-Reasoner under a self-play paradigm, we have conducted preliminary experiments in the Kuhn Poker environment within the limited rebuttal period. The performance results are listed as follows:
> | env | ours | SPIRAL | MARSHAL
> |--|--|--|--|
> |vs. KuhnPoker NE.Bot | 0.065 / -0.031 | -0.301 / -0.149 | -0.107 / -0.103 |
> |vs. LeducHoldem NE.Bot | -0.442 / -0.360 | -0.870 / -0.706 | -0.460 / -0.327 |
>
> As the results indicate, constrained to a self-play setting, our method **still maintains a clear performance advantage** over self-play baselines like MARSHAL and SPIRAL. It empirically demonstrates that our method is **effective in both shared and non-shared parameter configurations**.
>
> ---
>
> > ## W3 & Q3: Computational overhead.
>
> > **Action**: We will add this analysis of Computation in Section 5.5.
>
> We appreciate your insightful suggestion. To address your question regarding the micro-rollout size, we have conducted further analysis on both **computational cost and resulting performance** and plotted a **Pareto trade-off curve** (see https://anonymous.4open.science/r/micro_rollout-AACF/micro_rollout_pareto_front.pdf):
>
> Computational Overhead: Micro-rollouts are processed entirely in parallel. Consequently, the **computational bottleneck depends on the concurrent capacity of our framework**. As shown in the figure, when the micro-rollout size exceeds 2 (yielding approximately 600 concurrent requests), the **computational overhead increases significantly**. This indicates that the framework's parallel capacity is saturated, leading to queuing delays.
>
> Performance Impact: Similar to GRPO, increasing the micro-rollout size reduces the variance of the estimated advantage, leading to more stable training signals.
>
> The empirical results indicate that a **micro-rollout size of 2 achieves the best trade-off**, corresponding to **the knee point of the Pareto Front**, and is therefore adopted.
>
> ---
>
> > ## W4: Scalability to complex scenarios.
>
> We appreciate your understanding of our use of existing environments.
>
> **Our method can be extended to N-agent settings by allowing the ego agent to autonomously select which specific opponent to reason about**.
>
> Due to space constraints, for discussions regarding complex environment scalability, please refer to our response to Reviewer **HuSA (W2 & Q1)**; for multi-agent extensibility, please refer to Reviewer **qD1Y (W4 & Q3)**. We apologize for any inconvenience this may cause.
>
> ---
>
> Thanks again for your strong support and valuable suggestions, which have helped further strengthen our paper.
>
> Best,
>
> The Authors

---

> > ### Author Rebuttal · Reviewer_mKFL · 2026-04-02
> >
> > Thank you for the detailed and thoughtful response. I appreciate the clarifications and additional explanations provided. They addressed my main concerns well, and I am happy to maintain my current score.

---

> > > ### Author Response · Authors · 2026-04-02
> > >
> > > Thank you for the recognition of our paper!  We are delighted to hear that our rebuttal addressed your concerns in general.

---

### Official Review · Reviewer_HuSA · 2026-02-27

**Soundness:** 3
**Presentation:** 3
**Significance:** 2
**Originality:** 2
**Overall Recommendation:** 3
**Confidence:** 4

**Summary:**

This paper introduces Strat-Reasoner, a reinforcement learning (RL) framework designed to improve the strategic reasoning abilities of large language models (LLMs) in multi-agent games. The authors propose a recursive reasoning paradigm for LLMs, drawing from cognitive hierarchy theory, that enables agents to explicitly consider other agents' reasoning processes. A centralized Chain-of-Thought (CoT) comparison module quantifies the alignment between an agent's predictions and opponents' reasoning, providing rich intermediate signals. In addition, a hybrid advantage estimation integrates CoT-based evaluations with return-based advantages for stable optimization. Experiments across a range of adversarial and cooperative two-agent games show that the approach significantly improves both in-domain and out-of-distribution performance compared to strong open-source and closed-source baselines, with detailed ablation and generalization studies included.

**Compliance With Llm Reviewing Policy:**

Affirmed.

**Final Justification:**

The rebuttal clarified several points, but it did not resolve my main concern. The paper still lacks empirical validation in more complex environments and beyond the two-agent setting, which limits the strength of its broader claims. Since these issues are central to the paper’s significance, my overall assessment does not change.

**Key Questions For Authors:**

### 1. Scalability to complex environments

The evaluated benchmarks are relatively small-scale and short-horizon. How does the proposed recursive CoT comparison and hybrid advantage mechanism perform in substantially more complex strategic environments (e.g., longer horizons, larger action spaces, richer partial observability)?

Do the authors have preliminary evidence or theoretical justification suggesting that the framework remains stable and effective when scaling beyond small benchmark games?

**If the authors can provide empirical evidence demonstrating that the proposed method maintains stability and achieves clearer performance advantages—particularly in more complex strategic environments or against strong closed-source models—I would be inclined to raise my score.**

### 2. Multi-agent extensibility

The current formulation is tailored to two-agent alternating games. How would the recursive reasoning block generalize to N-agent settings where multiple agents must simultaneously model each other?

Would centralized CoT comparison require pairwise reasoning alignment across agents, and if so, how would computational and variance complexity scale?

Are there reasons to believe that the hybrid advantage estimation remains well-behaved under multi-agent belief hierarchies?

---

### 3. Prompt vs algorithm contribution

To what extent are the observed performance gains attributable to the structured recursive prompting itself versus the hybrid RL optimization mechanism?

Have the authors evaluated a structured-prompt-only baseline without CoT-based reward shaping to disentangle the contribution of prompting from the proposed training algorithm?

**Limitations:**

yes

**Strengths And Weaknesses:**

### Strengths

1. **Clear and interpretable reasoning structure**
   The proposed recursive reasoning block (illustrated in Figure 3 and operationalized in the case study) provides a distinctive and actionable interpretability advantage over opaque RL/self-play baselines. The structured decomposition of reasoning steps improves transparency and makes the strategic decision process easier to analyze and debug.

2. **Well-documented case study and prompt design**
   The explicit presentation of structured prompts and step-by-step reasoning traces (Appendices D and G) substantially enhances reproducibility. By exposing the reasoning templates and agent scaffolding, the paper enables further inspection and extension by future work.

---

### Weaknesses

1. **Incomplete engagement with related work on strategic LLM agents**
The related work section could further situate the contribution within recent LLM-based strategic decision-making research in more complex multi-agent environments, such as Diplomacy [1] and social deduction games like Werewolf [2]. Brief discussion of these directions would help clarify the positioning of the proposed framework within the broader landscape of strategic LLM agents.

2. **Limited task complexity weakens empirical claims**
   All evaluated environments (Tic-Tac-Toe, Kuhn Poker, MiniHanabi) are relatively small-scale and short-horizon benchmarks. In such settings, general-purpose closed-source models already achieve competitive performance without task-specific training. It remains unclear whether the proposed framework would demonstrate stronger advantages in more complex, long-horizon environments where stability and strategic consistency are more challenging.

3. **Limited validation of multi-agent extensibility**
   The experiments are restricted to two-agent alternating games. Although the paper claims extensibility to N-agent settings, no empirical validation (even in a simple three-agent setting) is provided. Given that recursive reasoning complexity increases substantially in multi-agent scenarios, additional evidence would be necessary to substantiate this claim.

### References

[1] *DipLLM: Fine-Tuning LLM for Strategic Decision-making in Diplomacy.*

[2] *Language Agents with Reinforcement Learning for Strategic Play in the Werewolf Game.*

---

> ### Author Rebuttal · Authors · 2026-03-29
>
> Hi Reviewer HuSA,
>
> We value your expertise and are delighted to submit our refined findings.
>
> ---
> > ## W1: Incomplete engagement with related work on strategic LLM agents.
>
> > **Action**: We will update our related work to include your suggested literature.
>
> We appreciate your suggestion and agree that incorporating references to complex multi-agent strategic environments, such as Diplomacy and social deduction games like Werewolf, is essential to better situate our work. We will update our related work to include your suggested literature. While recent research has explored complex environments primarily through training-free prompt engineering or game-specific training design, **our work takes a different approach by fundamentally optimizing the intrinsic strategic reasoning of LLMs via end-to-end RL**.
>
> ---
> > ## W2 & Q1: Scalability to complex environments.
>
> > **Action**: We will add this analysis to Section 5.
>
> We completely agree that transferring the framework to more complex environments is an exciting direction. However, empirical validation in such cases is currently bottlenecked by the **lack of standard benchmarks and relevant baselines (e.g., our baselines MARSHAL [ICLR 2026] and SPIRAL [ICLR 2026] both evaluate on these exact same short-horizon two-agent games)**.
>
> Nevertheless, in substantially more complex or long-horizon environments, **our method is arguably more advantageous compared to current baseline than it does in simple games**:
>
> **Stepwise Dense Signals for Long Horizons:** Our Centralized CoT Comparison provides a dense, stepwise training signal at every turn, directly mitigating the severe credit assignment problem inherent in long trajectories.
>
> **Handling Larger Action Spaces:** We do not strictly restrict the specific textual content of the reasoning fields. Therefore, when facing a larger action space, **the intent may adapt through more complex descriptions.** Furthermore, our scoring mechanism provides continuous scores, which can adapt to the complexity introduced by a larger action space.
>
> **Robustness via Normalization and Hyperparameter Control:** In exceptionally difficult environments where an agent's prediction accuracy might be uniformly low, GRPO normalization ensures the framework **still extracts an effective relative supervision signal.** Additionally, the CoT advantage weight ω can be dynamically tuned to balance the reasoning signal with environmental returns, ensuring training reliability across varying task difficulties.
>
> ---
> > ## W3 & Q2: Multi-agent extensibility.
>
> > **Action**: We will include the scalability analysis in the Appendix.
>
> Thank you for highlighting the importance of extensibility. We completely agree that naive pairwise reasoning (modeling all other agents) would drastically increase complexity.
>
> **Instead, we adopt a targeted reasoning pattern where we simply allow the ego agent to autonomously select its reasoning targets by adding a target-specification field.**  The centralized CoT comparison module can then retrieve ground-truths `MyIntent` and `MyPrediction` from explicitly specified opponents.
>
> Under this targeted reasoning formulation, the multi-agent belief hierarchy extends gracefully. The turn-level CoT Score simply averages the alignment scores of the selected targets (the averaging operation ensures that as the number of prediction items increases, the variance actually decreases, given that the correlation coefficient between prediction items is in $[0, 1)$). Combined with reasons mentioned in the answers above (W2 & Q1), it is reasonable to conclude that the hybrid advantage estimation remains highly reliable.
>
> ---
> > ## Q3: Prompt vs. algorithm contribution.
>
> > **Action**: We will revise "w/o CoT-adv" to "Prompt-only" in ablation studies for better clarity.
>
> We appreciate your insightful suggestion to disentangle the contributions of the prompting module and the RL algorithm. We completely agree, and **we have indeed evaluated a structured-prompt-only baseline in our ablation study (Section 5.3)**.
>
> In Figure 4, the gray dashed line initially labeled "w/o CoT-adv" represents this exact baseline—training exclusively with the recursive prompt module while excluding CoT training signals. As the curve illustrates, this baseline fails to yield sustained performance improvements, demonstrating that **prompting-only cannot intrinsically enhance the LLM's strategic reasoning capabilities**. In stark contrast, **our full method maintains a robust and significant upward trajectory**, substantiating that both the prompt and hybrid RL mechanism contribute to improving the LLM's strategic reasoning capability.
>
> We acknowledge that the original legend "w/o CoT-adv" might be unintuitive. Following your feedback, we will revise it to "Prompt-only" in the updated manuscript for better clarity.
>
> ---
>
> We appreciate your guidance and hope the updates meet your expectations.
>
> Best,
>
> The Authors

---

> > ### Author Rebuttal · Reviewer_HuSA · 2026-04-04
> >
> > We understand the authors’ explanation regarding the lack of benchmarks and baselines. However, we still believe that evaluations in more complex environments and with a larger number of agents are crucial and would significantly strengthen the study. Given the absence of such evidence, we prefer not to increase our score at this stage.

---

> > > ### Author Response · Authors · 2026-04-05
> > >
> > > Dear Reviewer HuSA,
> > >
> > > Thank you for your continued engagement and constructive feedback. We fully understand and respect your perspective regarding the evaluation in more complex and N-agent environments.
> > >
> > > ---
> > >
> > > We completely agree that evaluating in N-agent and more complex environments is a crucial future direction. **In fact, our design is inherently motivated by the scalability challenges encountered in such complex scenarios.** However, as it stands, rigorous empirical validation in N-agent settings is heavily bottlenecked by the lack of standardized benchmarks and applicable baselines. In the current LLM-based MARL domain, existing works either focus on using multi-agent systems to solve single-agent tasks (e.g., MAGRPO [1], MAPoRL [2]) or are evaluated exclusively in two-agent settings (e.g., our baselines MARSHAL [3] and SPIRAL [4]). **Therefore, our focus on foundational 2-player interactions actually highlights the core efficacy of our approach within the boundaries of current standard evaluation protocols, rather than limiting its potential.**
> > >
> > > We would like to emphasize that **the evaluated two-player setting already rigorously embodies the most critical challenges of multi-agent systems: the non-stationarity of other agents, the lack of effective signals guiding the reasoning process, and the multi-turn credit assignment problem.** When scaling to N-agent scenarios, the primary new obstacle is the increase in computational overhead caused by additional opponents. While naive pairwise modelling may cause exponential computational overhead, as you have correctly pointed out, our proposed "targeted reasoning" mechanism is a lightweight adaptation to directly address this scalability issue, without altering the core modules of our current method. **While a full-scale N-agent empirical comparison is currently pending the development of appropriate community benchmarks, our formulation conceptually avoids intractable complexity, laying a solid groundwork for N-agent extensions.**
> > >
> > > Ultimately, **we believe our work takes an important step forward by demonstrating how to generate reliable, turn-level training signals without the need for a specifically trained reward model, thereby effectively guiding an LLM's strategic reasoning. We provide a novel solution to the typical complex scenario of multi-agent multi-turn interactions through the lens of LLM reasoning.**
> > >
> > > We hope this clarifies the scope and foundational contribution of our current work. We genuinely appreciate your insightful perspective, which will undoubtedly guide our future research in N-agent extensions.
> > >
> > > ---
> > >
> > > References:
> > >
> > > [1] Liu S, Liang Z, Lyu X, et al. Llm collaboration with multi-agent reinforcement learning[C]//Proceedings of the AAAI Conference on Artificial Intelligence. 2026, 40(38): 32150-32158.
> > >
> > > [2] Park C, Han S, Guo X, et al. Maporl: Multi-agent post-co-training for collaborative large language models with reinforcement learning[C]//Proceedings of the 63rd Annual Meeting of the Association for Computational Linguistics (Volume 1: Long Papers). 2025: 30215-30248.
> > >
> > > [3] Yuan H, Xu Z, Tan Z, et al. MARSHAL: Incentivizing Multi-Agent Reasoning via Self-Play with Strategic LLMs[J]. The Fourteenth International Conference on Learning Representations. 2026. https://openreview.net/forum?id=GCd5v3ehmr
> > >
> > > [4] Liu B, Guertler L, Yu S, et al. Spiral: Self-play on zero-sum games incentivizes reasoning via multi-agent multi-turn reinforcement learning[J]. The Fourteenth International Conference on Learning Representations. 2026. https://openreview.net/forum?id=7Yayy5fNLg
> > >
> > > ---
> > >
> > > Best,
> > >
> > > The Authors

---

### Official Review · Reviewer_qD1Y · 2026-03-11

**Soundness:** 3
**Presentation:** 3
**Significance:** 3
**Originality:** 3
**Overall Recommendation:** 4
**Confidence:** 3

**Summary:**

The authors point out the challenge of non-stationarity of other agents in multi-agent games. To address this challenge, they propose Strat-Reasoner where an agent’s reasoning incorporates other agents’ reasoning. They also introduce a CoT comparison module for reasoning evaluation and improve LLMs’ performance in multi-agent games.

**Compliance With Llm Reviewing Policy:**

Affirmed.

**Final Justification:**

I hope the new results in author's response can be incorporated into the revised version of the paper. As such, I have raised my score from 3 to 4.

**Key Questions For Authors:**

-Why are micro-rollout size and CoT advantage weight set at particular values in Table5? How sensitive is the model performance to these hyperparameter values?

-Why did you choose the three specific environments? Are there any other environments that could be used to evaluate your method?

-What will change when the framework is extended to N-agent settings compared to a two-agent setting?

-Did you perform ablation for two other games other than MiniHanabi?

**Limitations:**

yes

**Strengths And Weaknesses:**

## Strength
-The paper is well motivated to incorporate the intuition “thinking about what others think”.

-The paper achieves performance improvement with proposed method and also provides resources to code to ensure reproducibility.

-The method is well explained with clear illustrations for each stage of the Strat-Reaoner framework and the recursive reasoning structure.

 -There is some generalization exploration by investigating out-of-distribution environments.

## Weakness
-Some key hyperparameters in the experiment like micro-rollout size and CoT advantage weight are fixed at a certain value without exploring other values, limiting the robustness of the method reported. It would be important to report performance by varying these hyperparameters to demonstrate the generalizability of the performance improvement brought about by the method.

-The author is using the base model as the judge model. It may introduce potential bias and reward hacking. Easier deployment may not be a strong reason considering the potential bias.

-The method was tested on three multi-agent environments. I’m concerned about the diversity and comprehensiveness of evaluation. As such, ‘significantly improve strategic abilities of models’ may be an overclaim.

-Although the authors talked about not exploring N-agent as an area for future work, it would be important to include this (at least to include one simple additional setting other than 2 agent only) since it can better prove the method’s effectiveness to multi-agent system.

-There is only one case study for Tic-Tac-Toe game. Since there are also different game settings, providing more examples (at least one representative of each game) would be beneficial.

---

> ### Author Rebuttal · Authors · 2026-03-29
>
> Hi Reviewer qD1Y,
>
> We truly appreciate your insightful feedback and are eager to share our updated findings.
>
> ---
>
> > ## W1 & Q1: Hyperparameter sensitivity analysis.
>
> **We set CoT advantage weight ω=0.2 to empirically balance unbiased, high-variance returns with dense, low-variance CoT signals.** As our sensitivity analysis shows, ω=0.3 may overfit local reasoning (degrading win rates), while ω<0.2 provides insufficient guidance. Thus, ω=0.2 optimally enhances reasoning as well as the primary game objective. Please see https://anonymous.4open.science/r/cot_adv-0377/minihanabi-cot-adv.pdf.
>
> We use micro-rollouts to establish a CoT advantage baseline. Experiments show **micro-rollout size M=2 achieves the best trade-off between performance gains and computational overhead (knee point of Pareto front)**. Please see https://anonymous.4open.science/r/micro_rollout-AACF/micro_rollout_pareto_front.pdf.
>
> ---
>
> > ## W2: Concerns about the judge model.
>
> > **Action**: We will add the analysis of the judge model in Appendix.
>
> In our method, the CoT comparison is explicitly disentangled into **independent semantic pairs**. As the agent, the base model performs recursive reasoning, while as the judge, it only evaluates semantic similarity between pairs, which is **independent of strategic reasoning and largely differs from open-ended subjective scoring**. Furthermore, the **contexts of the two tasks do not overlap**. Thus using the base model as the judge **does not introduce systematic bias or self-alignment effects**.
>
> Nevertheless, we agree that empirically validating the robustness of the training signal under different judge models is valuable. We **benchmarked the scoring performance across various models**. An example is shown below:
>
> Evaluated field: opponent_intent
>
> Prediction: Prevent opponent’s diagonal
>
> Ground Truth: placing X in (0, 2) to block diagonal
>
> | model | Qwen3-4B | Qwen3-32B | Deepseekv3.1 | GPT-5 | Claude4.5 | Gemini3.1 |
> |--|--|--|--|--|--|--|
> | score | 0.9 | 1.0 | 0.85 | 0.95 | 0.9 | 0.9 |
>
> As demonstrated, **other models yield highly consistent scores with Qwen3-4B. Consequently, we opted for the base model to optimize computational resources and deployment efficiency.**
>
> ---
>
> > ## W3 & Q2: Limited evaluation diversity.
>
> > **Action**: We will add this analysis in Appendix B.
>
> We selected these specific environments because they **systematically cover the core game-theoretic quadrants: perfect/imperfect information and competitive/cooperative dynamics**. Combined with **3 different OOD environments**, they rigorously **validate our method's generalizability.**
> While our framework can theoretically extend to broader multi-agent scenarios, available environments are limited. Therefore, we prioritized these highly representative games to prove our concept. Extending Strat-Reasoner to larger-scale environments remains a key focus for our future work.
>
> ---
>
> > ## W4 & Q3: Multi-agent extensibility.
>
> > **Action**: We will add this analysis in Appendix B.
>
> We completely agree that extending to N-agent settings is a vital direction. In the current LLM-based MARL domain, existing works either focus on using multi-agent systems to solve single-agent tasks (e.g., math reasoning in MAGRPO or MAPoRL) or are evaluated in two-agent settings (e.g., our baselines MARSHAL and SPIRAL). Thus, constructing a full N-agent setting empirically is currently constrained by the **lack of standard environments and comparable baselines**.
>
> However, our framework is **structurally capable of extending to N-agent settings**.
>
> For N-agent or simultaneous settings, **we allow the ego agent to autonomously select which specific opponent to reason about** within its `OpponentIntent` and `OpponentPrediction` fields (in alternating games, select targets from several previous turns; in simultaneous-move, select from exactly turn $t-1$), by adding a target-specification field. The centralized CoT comparison module can then retrieve ground-truths `MyIntent` and `MyPrediction` from explicitly specified opponents.
>
> In this way, the framework's logic **holds perfectly without altering its core mechanism.**
>
> ---
>
> > ## Q4: Ablation studies on other games.
>
> > **Action**: We will include other games' ablation in Appendix.
>
> Due to the limited rebuttal period, we prioritized conducting the ablation study on TicTacToe (a perfect-information competitive game). The results perfectly align with our Hanabi findings: **the full Strat-Reasoner achieves the highest return, while omitting CoT-adv degrades performance, and unnormalized CoT-scores cause instability. This confirms our framework's robustness across diverse game types.** Please see in https://anonymous.4open.science/r/ablation-2028/tictactoe-ablation.pdf
>
> > ## W5: Insufficient case studies.
>
> > **Action**: We will supplement case studies covering Kuhn Poker and Hanabi in Appendix G.
>
> ---
>
> Thank you again, we look forward to your thoughts on these improvements.
>
> Best,
>
> The Authors

---

> > ### Author Rebuttal · Reviewer_qD1Y · 2026-04-03
> >
> > I hope all of these points (rebuttal) will be incorporated into the revised version of the paper, as this is very important and necessary. I have raised my score.

---

> > > ### Author Response · Authors · 2026-04-03
> > >
> > > We are highly grateful for your increased score! We are fully committed to incorporating all these points discussed during this rebuttal phase into our revised paper.

---

### Official Review · Reviewer_p36w · 2026-03-13

**Soundness:** 3
**Presentation:** 3
**Significance:** 3
**Originality:** 2
**Overall Recommendation:** 4
**Confidence:** 4

**Summary:**

This paper aims to improve LLMs' strategic reasoning ability in multi-agent games. It introduces three modules, Recursive Reasoning, Centralized CoT Comparison, and Hybrid Advantage Estimation, which are progressively structured and interconnected. The essence of the approach is to encourage the LLMs to effectively model their opponents during the interaction process. The experimental evaluation is conducted on three games: Tic-Tac-Toe, Kuhn Poker, and MiniHanabi, and the results demonstrate the effectiveness of the proposed method.

**Compliance With Llm Reviewing Policy:**

Affirmed.

**Final Justification:**

I hope all of these points (rebuttal) will be incorporated into the revised version of the paper, as this is very important and necessary.
I have raised my score from 3 to 4.

**Key Questions For Authors:**

As one of the more important hyperparameters in this paper, how was w (Equation 9) determined? Were any ablation experiments conducted to explore the effect of different values of this hyperparameter?

**Limitations:**

See Weakness and Key questions.

**Strengths And Weaknesses:**

- This paper effectively translates the idea of opponent modeling into a reward signal within the RL process.
- It introduces Centralized CoT Comparison to encourage the model to perform opponent modeling correctly.
- The effectiveness of the proposed method is validated in both in-domain and out-of-domain (OOD) game scenarios.
- Recursive reasoning is an idea that has already been proposed in multiple prior works, such as [1], which consequently limits the novelty of the method.
- The training details need to be elaborated more carefully. Additionally, are the results presented in Tables 1 and 2 evaluated by running the trained model in the game environment? How was the trained model obtained (i.e., how was training stopped)?


[1]Zhang Y, Mao S, Ge T, et al. K-Level Reasoning: Establishing Higher Order Beliefs in Large Language Models for Strategic Reasoning[C]//Proceedings of the 2025 Conference of the Nations of the Americas Chapter of the Association for Computational Linguistics: Human Language Technologies (Volume 1: Long Papers). 2025: 7212-7234.

---

> ### Author Rebuttal · Authors · 2026-03-29
>
> Hi Reviewer p36w,
>
> Thank you for your valuable time. We appreciate your thorough review and are excited to share our updated findings with you.
>
> ---
> > ## W1: Related works about recursive reasoning limit the novelty of the method.
>
> > **Action:** We will update our related work to include your suggested literature.
>
> We appreciate your insightful suggestion. **Recursive reasoning originates from game theory [1]**, with existing works extending it to MARL [2], yet it remains underutilized in the domain of LLM-based agents. Although K-Level Reasoning (K-R [3]) has incorporated the concept of recursive thinking, **such approaches predominantly focus on designing prompting frameworks**, failing to fundamentally enhance the intrinsic strategic reasoning capabilities of LLMs.
>
> **Our proposed Strat-Reasoner leverages a recursive reasoning design to provide fine-grained training signals** for the agent's CoT, addressing the sparse reward problem inherent in long-horizon decision-making, and **systematically enhancing the strategic reasoning capabilities of LLMs through reinforcement learning**.
>
> ---
>
> > ## W2: Training details, in-game evaluations for Tables 1 & 2, and stopping criteria.
>
> > **Action**: We will update the training details in Appendix C.
>
> We appreciate your valuable feedback, which helps to improve our paper.
>
> **Training details:**
>
> Strat-Reasoner is implemented using the **ROLL [4] framework and OpenSpiel [5] for game integration**, fine-tuning the Qwen3-4B model via reinforcement learning with LoRA. More details can be found in Appendices C and F. To briefly elaborate, **our framework employs a strict on-policy RL strategy**. Two distinct LLM agents interact with each other to generate trajectories and then fine-tuned **respectively** on their own trajectories data.
>
> **Evaluation in the Game Environment:**
>
> **Yes, all results in Tables 1 & 2 are dynamically evaluated by running trained models in the OpenSpiel environment, not on static datasets**. For example, the 90.77 in Table 1's Tic-Tac-Toe column denotes the normalized test return of our trained model evaluated against the MCTS-100sims bot.
>
> **Stopping Criteria (How the model was obtained):**
>
> Following recent LLM-MARL works [6, 7], we employ an **online validation mechanism**. Every 10 steps, the model is evaluated against a static bot (e.g., MCTS-100sims bot). **Training halts when performance converges or drops (Early Stopping)**. The final evaluation results reported in Tables 1 and 2 were generated using the last checkpoint saved upon the completion of training.
>
> ---
>
> > ## Q1: How was Cot advantage weight ($\omega$) determined?
>
> > **Action**: We will add the empirical analysis of $\omega$ in Appendix.
>
> Thank you for the insightful comments. As you correctly pointed out, the CoT advantage weight $\omega$ plays a crucial role in balancing our optimization objectives. **The determination of $\omega=0.2$ was achieved through empirical ablation studies, guided by the theoretical differences in signal properties.** In our framework, the outcome-based signal (Equation 8) relies on a return-based advantage with a trajectory-level baseline. This signal is unbiased but suffers from high variance and sparsity. In contrast, our CoT advantage operates similarly to GRPO, utilizing micro-rollouts to provide a dense, low-variance signal normalized to a standard distribution.
>
> Because of this discrepancy in signal density and variance scale, applying equal weight would risk overwhelming the objective with immediate reasoning signals, potentially diluting the ultimate goal of the game. **We conducted a sensitivity analysis on $\omega$ in the hanabi environment**, and the newly added figure illustrates these findings. You can find it at https://anonymous.4open.science/r/cot_adv-0377/minihanabi-cot-adv.pdf
>
> As shown in the figure, $\omega < 0.2$ yields a CoT signal too weak to guide reasoning, limiting performance gains. Conversely, $\omega > 0.2$ may cause the model to overfit to local CoT alignment by perfectly predicting the opponent's current turn, which comes at the expense of long-term strategy and the ultimate win rate. **Thus, $\omega=0.2$ serves as the empirical sweet spot, balancing the local CoT training signal derived from recursive reasoning with final game outcomes**.
>
> ---
>
> References:
>
> [1] Camerer et al. A cognitive hierarchy model of games[J]. Quart.J.Econ. 2004.
>
> [2] Wen et al. Probabilistic recursive reasoning... arXiv:1901.09207 (2019).
>
> [3] Zhang et al. K-Level Reasoning... NAACL 2025.
>
> [4] Wang et al. Reinforcement learning optimization for large-scale learning... arXiv:2506.06122 (2025).
>
> [5] Lanctot et al. OpenSpiel: A framework... arXiv:1908.09453 (2019).
>
> [6] Yuan et al. Marshal... arXiv:2510.15414 (2025).
>
> [7] Liu et al. Spiral... arXiv:2506.24119 (2025).
>
> ---
>
> Thank you again for your valuable time and insightful feedback. We appreciate your guidance and hope the updates meet your needs.
>
> Best,
>
> The Authors

---

> > ### Author Rebuttal · Reviewer_p36w · 2026-04-03
> >
> > Thank you for the detailed response. I hope all of these points will be incorporated into the revised version of the paper, as this is very important and necessary. I have raised my score.

---

> > > ### Author Response · Authors · 2026-04-03
> > >
> > > Thank you so much for the score increase! We ensure that all these points provided during this rebuttal will be carefully integrated into the revised paper.

---

### Decision · Program_Chairs · 2026-04-30

**Decision:**

Accept (regular)

**Comment:**

This paper proposes Strat-Reasoner, an RL-based framework aimed at improving LLMs’ strategic reasoning in multi-agent games by introducing recursive reasoning structures, centralized CoT comparison, and hybrid advantage estimation.

Reviewers generally find the problem important and timely, particularly in addressing non-stationarity and strategic interaction in multi-agent settings, and note that the framework is well-structured with interpretable components. The empirical results across several games show consistent improvements over baselines, and the inclusion of detailed analyses and resources supports reproducibility.

While some reviewers raise concerns about limited novelty relative to existing recursive reasoning and MARL approaches, as well as restricted evaluation to relatively simple environments, others find the technical integration and empirical validation sufficiently solid for publication. Concerns regarding scalability, hyperparameter sensitivity, and reliance on a judge model are valid, but the authors’ rebuttal provides reasonable clarifications and outlines concrete steps for addressing these issues. In particular, the methodological design and observed gains suggest the approach captures meaningful aspects of strategic reasoning beyond standard baselines.

Overall, despite some limitations in evaluation breadth and positioning, the paper makes a meaningful contribution to an emerging area at the intersection of LLMs and multi-agent reasoning. I therefore recommend a weak accept, as the strengths modestly outweigh the weaknesses and the work is likely to stimulate further research in this direction.